# FDA-ARGOS is a database with public quality-controlled reference genomes for diagnostic use and regulatory science

Heike Sichtig [1], Timothy Minogue[2], Yi Yan[1], Christopher Stefan[2], Adrienne Hall[2], Luke Tallon[3], Lisa Sadzewicz[3], Suvarna Nadendla [3], William Klimke[4], Eneida Hatcher[4], Martin Shumway [4], Dayanara Lebron Aldea [5], Jonathan Allen [5], Jeffrey Koehler[2], Tom Slezak[5], Stephen Lovell[1], Randal Schoepp [2] & Uwe Scherf [1]

FDA proactively invests in tools to support innovation of emerging technologies, such as infectious disease next generation sequencing (ID-NGS). Here, we introduce FDA-ARGOS quality-controlled reference genomes as a public database for diagnostic purposes and demonstrate its utility on the example of two use cases. We provide quality control metrics for the FDA-ARGOS genomic database resource and outline the need for genome quality gap filling in the public domain. In the first use case, we show more accurate microbial identification of *Enterococcus avium* from metagenomic samples with FDA-ARGOS reference genomes compared to non-curated GenBank genomes. In the second use case, we demonstrate the utility of FDA-ARGOS reference genomes for Ebola virus target sequence comparison as part of a composite validation strategy for ID-NGS diagnostic tests. The use of FDA-ARGOS as an in silico target sequence comparator tool combined with representative clinical testing could reduce the burden for completing ID-NGS clinical trials.

[1] U.S. Food and Drug Administration, 10903 New Hampshire Ave, Silver Spring, MD 20993, USA. [2] U.S. Army Medical Research Institute of Infectious Diseases, 1425 Porter Street, Frederick, MD 21702, USA. [3] Institute for Genome Sciences at the University of Maryland, 670 W. Baltimore Street, Baltimore, MD 21201, USA. [4] National Center for Biotechnology Information, National Library of Medicine, 8600 Rockville Pike, Bethesda, MD 20894, USA. [5] Lawrence Livermore National Laboratory, P.O. Box 808, Livermore, CA 94551, USA. Correspondence and requests for materials should be addressed to H.S. (email: Heike.Sichtig@fda.hhs.gov) or to T.M. (email: Timothy.D.Minogue.civ@mail.mil)

Patients and clinicians need alternative solutions when conventional diagnostics (e.g., real-time PCR, culture or ELISA) fail to identify an infectious etiology. Several studies document this need of applying hypothesis-free NGS as a diagnostic of last resort, such as high-risk transplant population or failure of diagnosis with conventional diagnostics[1,2]. Numerous groups have successfully applied ID-NGS technology across several unique and diverse clinical use cases. For example, isolate shotgun sequencing information uncovered unexpected transmission routes during multi-drug resistant nosocomial organism outbreaks[3–5]. Other studies showed use of targeted sequencing to group *E. coli* clonotypes from patient's direct urine samples[6], or to detect ciprofloxacin resistance markers[7], resulting in antimicrobial susceptibility data and improvement in clinical outcome prediction. Finally, agnostic (unbiased, metagenomic) sequencing shows promise as a diagnostic of last resort where no other diagnostic can determine the infectious microorganism, such as the successful ID-NGS diagnosis of leptospira infection with resulting positive outcome for the patient[8].

Infectious disease next generation sequencing (ID-NGS) diagnostics, with the potential to identify any microbial organism or genomic marker from a patient sample in a single test, are poised to enter the clinical diagnostic laboratory[9–11]. ID-NGS is finding application across the infectious disease space; however, several studies document the continued need for NGS research and database curation to facilitate adoption in the clinical setting[2]. Perhaps the best example, Afshinnekoo et al. showed ID-NGS misidentification of anthrax and plague in the NYC subway system based on low-quality reference genomes[12]. A follow-up erratum by the same group[13] revealed the lack of evidence for biothreat organisms in these samples. This erratum attributed the anthrax misidentification to poor reference genomes leading to misattribution of toxin genes when using metagenomic data analysis tools. This lack of proper reference genomes is pervasive and represents significant knowledge gaps in public resources, thus emphasizing the necessity for targeted development of representative, accurate and well curated microbial reference genome sequences. Additional studies showed that effective use of agnostic sequencing technology, either for infectious disease identification or exclusion of infectious etiologies, is directly related to the availability of quality-controlled whole-genome reference sequences[14–16]. Significant efforts are still required for ID-NGS technology to transition into a routine clinical diagnostic. To facilitate this transition, prominent groups and researchers in the field have outlined steps required for proper ID-NGS use in the clinic[2,17,18].

This manuscript provides our rationale and quality metrics for the FDA-ARGOS reference genome database to build NGS infrastructure and outlines the need for genome gap filling in the public domain. Furthermore, we demonstrate the utility of the FDA-ARGOS database on the example of two use cases: (1) accurate identification of *E. avium* from metagenomic samples and (2) target sequence comparison in combination with representative clinical testing as a composite reference method (Fig. 1) for ID-NGS diagnostics.

## Results

**Filling targeted genome gaps in public resources.** In 2013, FDA in collaboration with the Department of Defense (DoD) and the National Center for Biotechnology Information (NCBI) assessed the quality and diversity of sequenced microbial genomes present in public databases. A majority of pathogens appeared to be represented by multiple entries, however, many of these genomes were incomplete or of unknown quality. In fact, a thorough examination of the entire public domain revealed some pathogens

were underrepresented or completely absent. Our 2013 review, supported by several publications[19–21], revealed biased phylogenetic coverage usually attributable to research funding for specific microbial model organisms. At the time, NCBI GenBank covered <8000 bacterial and archaeal genome sequences with at least half submitted by the four largest genome sequencing centers: Broad Institute, DOE Joint Genome Institute, Institute for Genome Sciences and TIGR/JCVI. In addition, many sequences lacked accompanying metadata and raw read information. These issues provided the impetus for de novo generation of FDA-sponsored reference sequences of the highest quality achievable using state-of-the-art genomic sequencing technologies[22]. With this effort, FDA intended to establish quality control metrics for microbial genomes that could be used for ID-NGS test validation. Only genomes with the highest technically achievable quality would qualify as regulatory-grade genomes. Factors essential to reach that goal were: (1) knowledge of the technology used to generate the sequences, (2) access to raw sequence information to reproduce the data, and (3) access to relevant metadata. Perhaps the most significant missing piece of information for previously generated reference genomes was the lack of an independent reference method that reliably linked the microbial organism identification to the sequence data. In this context, qualification of microbial reference genomes requires organism identification with a recognized reference method as this remains a primary requirement for validation of a diagnostic device.

FDA, DOD, NCBI and other agencies identified more than 1000 diagnostic relevant high-quality genome gaps in public microbial genomic repositories using scientific literature, a phylogenetic data mining approach, and FDA microbial species-specific guidance documents. We prioritized these genome gaps and selected diagnostic relevant microorganisms, including biothreat microorganisms, common clinical pathogens and near-neighbor species (https://argos.igs.umaryland.edu/doc/pdf/wanted-orgnaism-list-Jan2019.pdf). The primary objective of this regulatory science research and tool development effort centered on the generation of an initial set of 2000 quality-controlled microbial FDA-ARGOS reference genomes. These genomes are generated with a hybrid assembly approach using short and long read sequencing technologies[22]. An initial collection criterion focused on sequencing at least five diverse isolates per species to cover temporal and spatial genome plasticity and initiate the construction of a regulatory-grade microbial genome model.

**Introducing reference genomes for diagnostic purposes.** FDA and collaborators established the publicly available database, FDA dAtabase for Regulatory-Grade micrObial Sequences (FDA-ARGOS), to fill these defined quality gaps for diagnostic relevant genomes. The goal of FDA-ARGOS is to generate and make publicly available near finished genomes of sufficient quality for diagnostic purposes. Here, we present the first subset of 487 FDA-ARGOS genomes with NCBI accessions (Fig. 2, Supplementary Data 1). Of the 487 isolates, 88.3% were bacteria, 11.1% were viruses, and 0.6% were eukaryotes, representing 189 different taxa. In total, 81.9% of genomes were of clinical origin with the remaining 18.1% environmental genomes from closely related species near-neighbors (Supplementary Table 1). Over 1000 isolates are currently being sequenced and at different stages in the FDA-ARGOS genome generation pipeline.

Use of advanced sequencing technologies[22] helped define the characteristics for regulatory-grade genomes for diagnostic use. Specifically, Fig. 1b provides a summary of required FDA-ARGOS metrics to support a determination of a regulatory-grade genome. All FDA-ARGOS genomic submissions demonstrated:

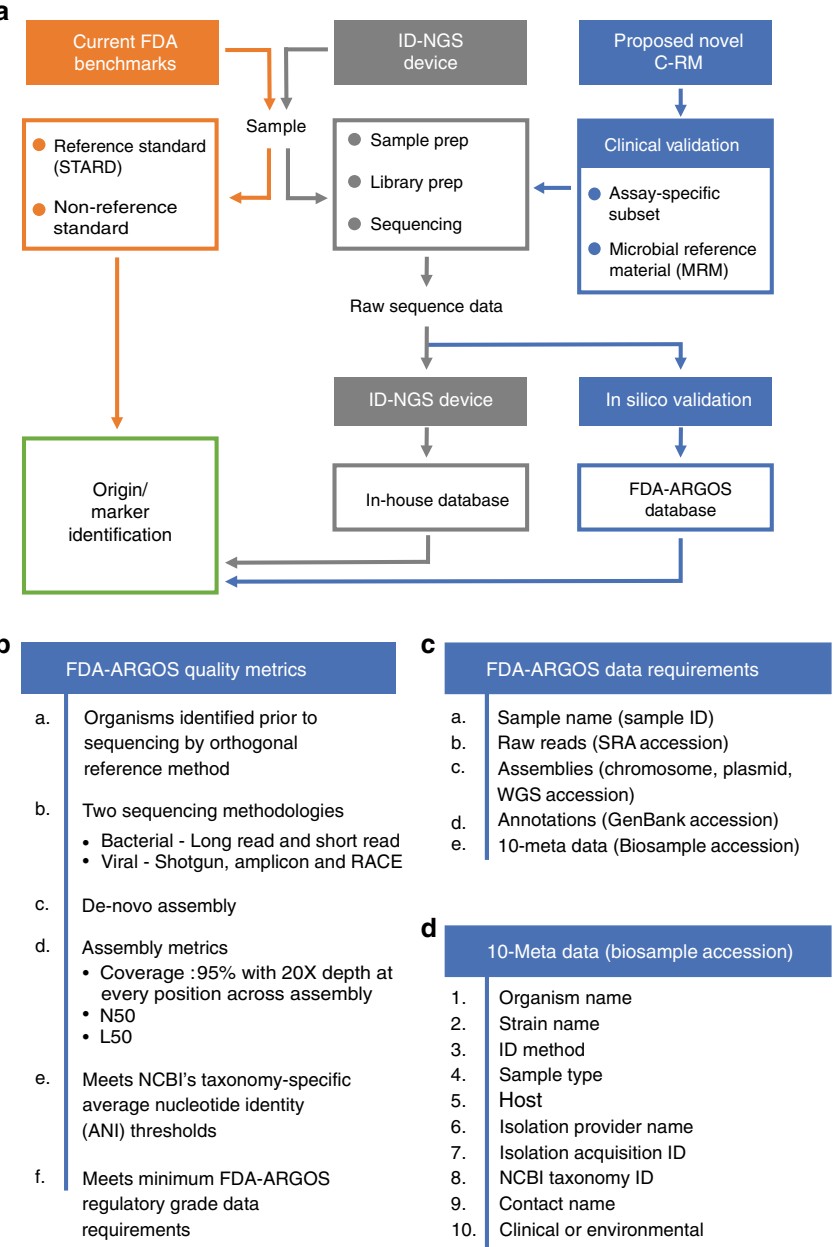

**Fig. 1** Proposed composite reference method (C-RM) for ID-NGS diagnostics. Panel **a** illustrates a walkthrough of the C-RM. Here, we show in silico target sequence comparison with FDA-ARGOS reference genomes in combination with representative clinical testing to understand the performance of ID-NGS diagnostic tests. Using raw sequence data from the ID-NGS diagnostic test device, in silico comparison of results obtained with the assay in-house database to results when using FDA-ARGOS will evaluate device bioinformatic analysis pipelines and report generation while eliminating the need for additional sample testing with a gold standard comparator (current FDA benchmarks). Overall, we anticipate the use of the C-RM based on assay-specific subsets of clinical samples and/or microbial reference materials (MRMs) for clinical validation in combination with FDA-ARGOS in silico target sequence comparison to generate scientifically valid evidence for understanding the performance of ID NGS diagnostic tests. Panel **b** lists the required quality control metrics for passing the regulatory-grade reference genome criteria. At a minimum, an FDA-ARGOS regulatory-grade reference genome adheres to six metrics (a–f). Specifically, category f details the minimum data requirements that are further described in (**c**). In addition, panel **d** lists the 10 critical metadata that need to be ascribed to a genome to meet the regulatory-grade criteria

(1) organism identification prior to sequencing by a recognized independent reference method, (2) sequence generation with at least two sequencing methodologies (e.g., long read and short read NGS), and (3) de novo assembly with high-depth of base coverage. Each microbial isolate assembled genome sequence conformed to a minimum of 95% coverage with 20× depth at every position while also providing concordant NCBI taxonomy-specific average nucleotide identity (ANI) thresholds for

microbial organism identification[23] with independent identification methods. All FDA-ARGOS samples were concordant between de novo sequencing identification and independent organism identification methods (see Supplementary Table 1).

As mentioned above, hybrid error correction with long and short read sequencing technology was considered for establishing minimum FDA-ARGOS regulatory-grade data requirements. Figure 1c outlined these criteria including: sample name, 10

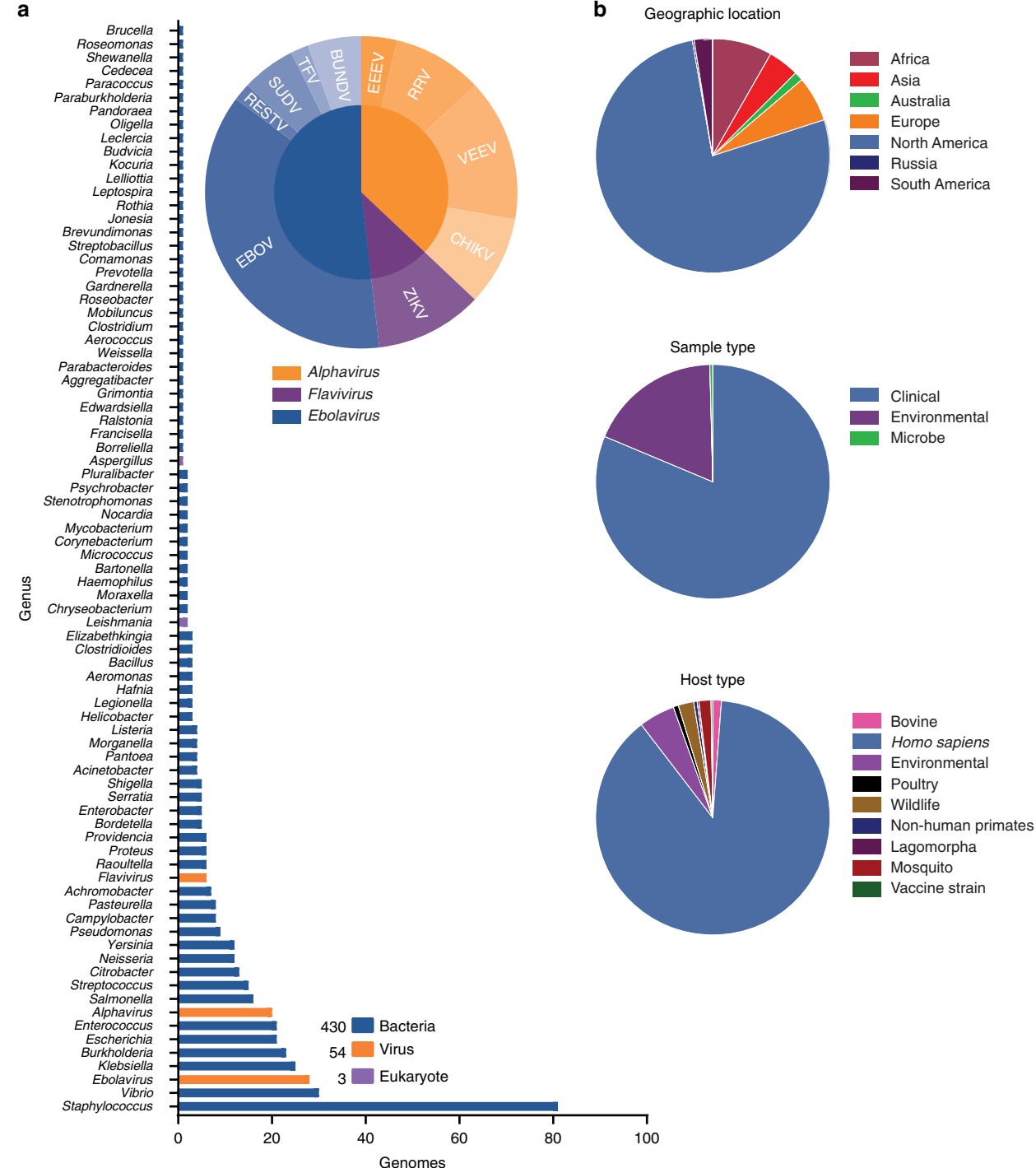

**Fig. 2** FDA-ARGOS quality-controlled reference genomes for diagnostic use. Summary statistics of the current 487 microbial genomes show primary coverage of FDA-ARGOS resides with bacterial isolates, followed by viruses and then eukaryotic parasites (**a**). Supplementary Data 1 provides accessions for all 487 genomes currently available publicly. A majority of FDA-ARGOS constituents (**b**) originate from North America and are from human clinical isolation

metadata fields (based on NCBI BioSample submission requirements), raw reads, assemblies with coverage, N50, L50, and annotations. Importantly, FDA-ARGOS genomes are tied to a minimum of 10 critical sample metadata fields (Fig. 1d): independent organism confirmation by recognized reference method, culture collection, and the following required NCBI BioSample fields: organism, strain, isolation source, host, collected by, taxonomy ID, contact, and package information.

Supplementary Table 1 shows metadata coverage metrics for all 487 FDA-ARGOS genomes. The 10 sample metadata fields are 100% completed and available throughout the sample set with five additional metadata metrics recommended, such as geographic location, collection date, host disease, host sex, and host age (https://www.ncbi.nlm.nih.gov/biosample/docs/attributes/). In terms of clinical representation, 81.9% of clinical samples in the collection are associated with known phenotype/host disease.

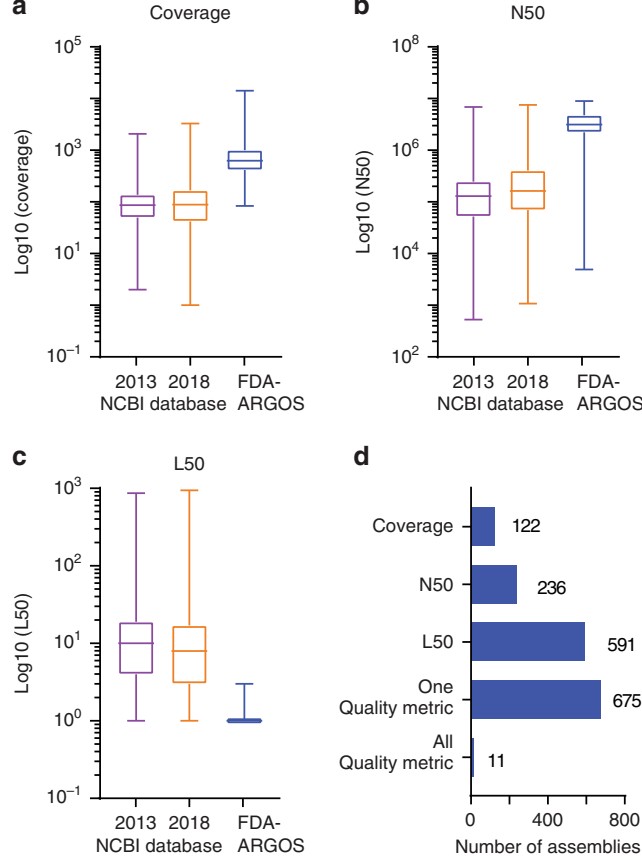

**Fig. 3** FDA-ARGOS reference genome assemblies quality metrics. Comparative microbial genome assembly quality metrics contrasted current FDA-ARGOS assemblies to 2013 and 2018 NCBI GenBank assemblies submitted for each species captured within the FDA-ARGOS database. Assembly quality metrics measured included: (**a**) median coverage, (**b**) median N50, (**c**) median L50, and (**d**) number of 2018 NCBI genomes that exhibited all, one or a specific quality control metric used to vet FDA-ARGOS genomes for inclusion. The NCBI assemblies were downloaded on August 6, 2018. For each box plot the center line represents the median value and is bounded by the 25th and 75th percentiles. The whiskers represent the min and max values

Critical for the designation as regulatory-grade genomes for diagnostic use, was the institution of quality control metrics for all aspects of the genome generation. To objectively identify such quality control metrics, we performed internal quality control assessments of all 487 genome assemblies (see Methods section for calculation of FDA-ARGOS genome assembly quality control statistics and Supplementary Data 1). Figure 3 shows the quality of FDA-ARGOS genome assemblies compared with the representative 2013 NCBI GenBank database and the representative 2018 NCBI GenBank database. Both, the 2013 and 2018 NCBI database captures held up to 50 NCBI assemblies for each species within the FDA-ARGOS database from the respective year. In relative number of assemblies, 2018 NCBI database contained 3535 while the 2013 contained 1617. Overall, we observed higher quality in the FDA-ARGOS genome dataset for the coverage, N50, and L50 quality assembly metrics compared with the 2013 and 2018 NCBI GenBank public genome dataset (Fig. 3a–c, respectively). Figure 3d demonstrated that only 675 out of the 3535 2018 NCBI GenBank assembled genomes, or 20%, showed comparative assembly quality to FDA-ARGOS genome sequences when considering one of the reported assembly quality metrics.

More importantly, when considering all quality control assembly metrics, only 11 out of the 3535 2018 NCBI GenBank assembled genomes, or 0.3%, showed comparable quality to FDA-ARGOS genome assemblies.

We expect refinement of the quality metrics for regulatory-grade genome status (Fig. 1b) as we continue to populate FDA-ARGOS with additional quality-controlled genomes; therefore, we established the requisite that all genomes should be available publicly. Deposition of all FDA-ARGOS genomes requires that raw reads, assembled genomes, and associative metadata are publicly available (https://www.ncbi.nlm.nih.gov/bioproject/231221) (check https://www.fda.gov/medical-devices/science-and-research-medical-devices/database-reference-grade-microbial-sequences-fda-argos for additional background information and updated genomes).

**Accurate identification of *E. avium* as use case 1**. Several regulatory science considerations arose during the process of generating FDA-ARGOS genomes, including the initial impetus for this effort, gap filling with diagnostic valuable quality-controlled reference genomes. Our first use case documented the importance of genome gap filling with FDA-ARGOS quality-controlled genomes, and the impact of lack of high-quality publicly available genomes for medically important microbes on potential diagnostic applications. Specifically, we tested whether the addition of quality-controlled reference sequences into the public repositories impacted the NGS pathogen detection of a metagenomic shotgun sequencing approach of a mock clinical *E. avium*-spiked human blood sample at clinically relevant titers. An isolate from reference genome SAMN04327393, which was removed from reference databases for data analysis, was used as a mock clinical *E. avium* sample. Initial read mapping using CLC Genomics and *E. avium* sequences from publicly available databases as a reference demonstrated de novo assembly of *E. avium* data was not possible due to only an average of 424.4 mapped paired-end reads (Supplementary Table 2). For frame-of-reference, we would need over 600,000 reads for de novo assembly of an entire *E. avium* genome of ~5 Mb at 20× coverage, assuming a read size of 150 bp and perfect quality of each generated read at all positions.

Metagenomic sequencing from clinical matrix resulted in low pathogen specific read counts and de novo assembled contigs, requiring diagnostic calls from reference coverage of <1×. Due to this lack of coverage, nucleotide errors in reference genomes can result in read misalignments and diagnostic false positives. To test this assertion, we generated 200 independent randomized database instances subsampling NCBI Nt or FDA-ARGOS assemblies (Supplementary Data 1). Each database instance had identical microorganism species composition and number of assemblies per species. Species composition was constructed from the empiric intersection of the FDA-ARGOS assemblies and NCBI Nt assemblies. Subsequently to determine this intersection, we ran 1200 simulations using MegaBLAST[24] or Kraken[25] with the same parameters, cutoffs and an identical number of randomly selected representative species assemblies from NCBI Nt and FDA-ARGOS and the triplicate *E. avium* DNA metagenomics samples. The normalized NCBI Nt and FDA-ARGOS database instances showed an average of 61 *E. avium* reads (minimum 18, maximum 109).

A detailed investigation for all mapped reads showed a clear picture of genome quality impact on accurate species identification (see Supplementary Data 3). Of the 1200 metagenome simulations, 600 simulations were performed with the MegaBLAST tool and normalized NCBI Nt or FDA-ARGOS database instances (see Supplementary Data 4). Specifically, while NCBI Nt yielded 19.26% total mapped reads versus a 0.007% for

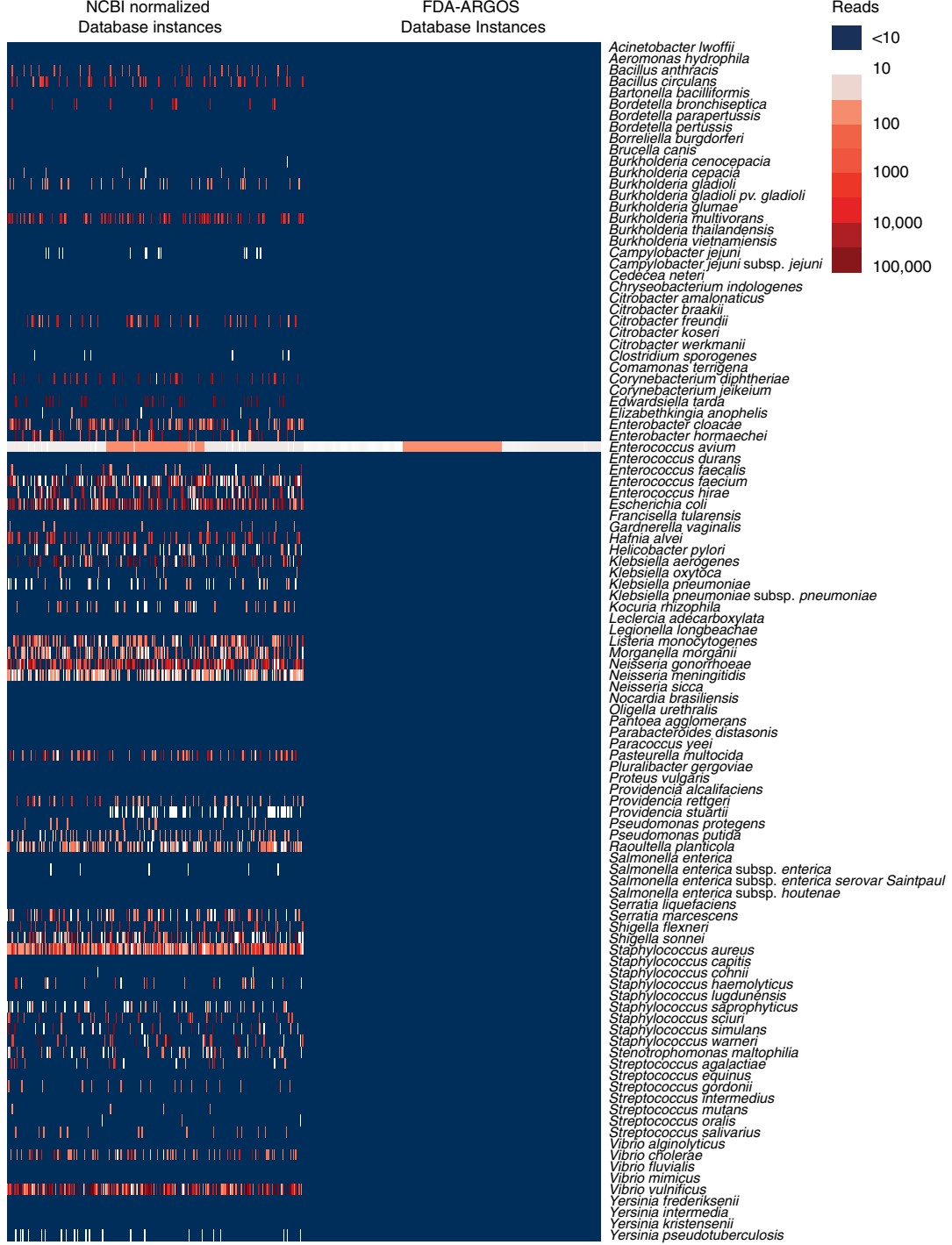

**Fig. 4** Comparison of NCBI Nt and FDA-ARGOS read classification results. Visualizing bioinformatics analysis with the MegaBLAST tool of metagenomics shotgun data of mock clinical human blood sample spiked with $10^5$ *E. avium*. The heatmap showed read classification results for triplicate samples run against 200 database instances. Dark blue indicates read numbers below 10. A gradient from white to red indicates read numbers ranging from above 10 to 100,000. Here we demonstrated read classification results for all simulated species. *E. avium* classification results were consistent across all database instances. In addition, several other species were classified at >1000 reads with the normalized NCBI Nt database instances (Supplementary Data 3 and 4)

FDA-ARGOS (Fig. 4), NCBI Nt mapped reads to over 30 different species other than the pathogen in this sample, *E. avium*. The highest number of reads, 34,879 reads on average, matched *Vibrio vulnificus* compared with 62 reads on average for *E. avium*. In contrast, MegaBLAST simulations with FDA-ARGOS yielded a single species, *E. avium*, with 60 reads on average. Further analysis of results obtained with NCBI Nt database instances and the MegaBLAST tool revealed mislabeled genomes due to human contamination. Kraken produced similar results (see Supplementary Data 5).

Finally, we performed *E. avium* isolate shotgun sequencing without clinical matrix to look at the impact of genome quality to remove any potential confounder-effect from sample matrix. We ran 1200 isolate simulations and all of them resulted in consistent

**Table 1 Bundibugyo ebolavirus performance summary**

| Sample | Real-time PCR (benchmark) $C_q$ value | MIPS (test device) % Classified | FDA-ARGOS (MegaBLAST) % Classified | FDA-ARGOS (Kraken) % Classified |
|---|---|---|---|---|
| 2012-1 | **22.97/22.95** | **54.95%** | **59.84%** | **70.89%** |
| 2012-16 | ND | 0.02% | 0.76% | 0.02% |
| 2012-91 | ND | 0.03% | 0.65% | 0.04% |
| 2012-95 | ND | 0.02% | 0.59% | 0.03% |
| 2012-99 | ND | 0.02% | 0.67% | 0.06% |
| 2012-120 | **23.46/23.38** | **41.87%** | **45.14%** | **57.56%** |
| 2012-147 | **25.58/25.52** | **27.23%** | **29.78%** | **47.52%** |
| 2012-153 | **28.14/27.96** | **38.30%** | **40.97%** | **50.70%** |
| 2012-176 | **37.01/36.54** | 0.01% | 0.87% | 0.01% |
| 2012-198 | ND | 0.02% | 0.71% | 0.03% |
| NTC | N/A | 0.02% | 1.59% | 1.05% |

Illustration of target sequence comparison with FDA-ARGOS reference genomes for diagnostic performance testing. This table shows the traditional benchmark comparison of the Bundibugyo MIPS assay to real-time PCR (RT-PCR) results and target sequence comparison with FDA-ARGOS using two bioinformatics tools (MegaBLAST and Kraken). Benchmark positive values were only noted for samples that yielded duplicative positive results by RT-PCR (bolded). Percent reads classified only refer to percentage of reads that were assigned to Bundibugyo ebolavirus, the remaining reads are non-specific

**Table 2 Zaire ebolavirus Makona performance summary**

| Sample | Real-time PCR (benchmark) $C_q$ value | MIPS (test device) % Classified | FDA-ARGOS (MegaBLAST) % Classified | FDA-ARGOS (Kraken) % Classified | FDA-ARGOS (LMAT) % Classified |
|---|---|---|---|---|---|
| 3754-2 | **35.11/35.72** | 0.05% | 0.60% | 0.06% | 0.03% |
| 3754-4 | **33.83/33.36** | 0.06% | 0.68% | 0.06% | 0.03% |
| 3811-2 | **36.17/36.14** | 0.07% | 0.56% | 0.08% | 0.04% |
| 3856-1P | **15.95/15.98** | 76.63% | 80.91% | 79.50% | 42.48% |
| 3913-5 | **34.00/33.77** | 0.00% | 0.68% | 0.00% | 0.01% |
| 3958-4 | **32.77/33.28** | 0.04% | 0.65% | 0.05% | 0.05% |
| 3991-2 | **33.92/33.62** | 0.00% | 0.65% | 0.00% | 0.01% |
| 4007-2 | **26.30/26.55** | 21.33% | 22.41% | 21.81% | 12.12% |
| 4015-1 | **21.66/21.66** | 74.87% | 76.35% | 75.82% | 39.35% |
| 4033-1 | **16.59/16.32** | 76.64% | 79.59% | 78.97% | 41.79% |
| 4268-1P | **25.05/25.15** | 29.71% | 30.43% | 29.97% | 15.87% |
| 4468-3 | **35.78/35.91** | 0.04% | 0.59% | 0.05% | 0.03% |
| 4641-3P | **31.81/31.82** | 0.03% | 0.59% | 0.04% | 0.02% |
| 4726-1 | **21.22/21.21** | 53.95% | 56.44% | 54.28% | 30.05% |
| 4845-3 | **35.17/36.71** | 0.00% | 0.66% | 0.01% | 0.01% |
| NTC | N/A | 0.02% | 0.86% | 0.04% | 0.01% |

Illustration of target sequence comparison with FDA-ARGOS reference genomes for diagnostic performance testing. This table shows the traditional benchmark comparison of the EBOV MIPS assay to real-time PCR (RT-PCR) results and target sequence comparison with FDA-ARGOS using three bioinformatics tools (MegaBLAST, Kraken, and LMAT). Benchmark positive values were only noted for samples that yielded duplicative positive results by RT-PCR (bolded). Percent reads classified only refer to percentage of reads that were assigned to Zaire ebolavirus Makona, the remaining reads are non-specific

*Enterococcus* genus calls with *E. avium* as the top species for both normalized NCBI Nt and FDA-ARGOS database instances. However, several simulations with the normalized NCBI Nt database showed *Enterococcus hirae* as top hit (Supplementary Data 6, 7, and 8). Further analysis of these results showed that one specific *Enterococcus hirae* genome was the top hit. The calculated ANI score for SAMN03198084 (Supplementary Data 2) showed higher correlation to the *E. avium* genomes than the remaining *Enterococcus hirae* genomes.

For future benchmarking efforts of bioinformatics tools, we provided all *E. avium* reference datasets and databases (see Data availability section: Reference Datasets a and b, and Supplementary Data 2).

**Ebola virus ID-NGS diagnostic C-RM validation as use case 2.** A major incentive for the development of FDA-ARGOS is to enable and promote innovation for ID-NGS medical devices. Through the process of populating the FDA-ARGOS database,

the concept of partial in silico validation, rather than completely empirical validation of clinical trial samples with an independent gold standard reference method, matured. We chose FDA-ARGOS Ebola virus reference sequences (Supplementary Data 1) and a targeted ID-NGS assay, Ebola virus molecular inversion probes (MIPS), to evaluate the application of FDA-ARGOS as an in silico target sequence comparison tool. Tables 1 and 2 show the diagnostic performance of the MIPS ID-NGS assay with clinical Bundibugyo virus and EBOV Makona samples reported as a more sensitive assay than the EBOV real-time PCR (RT-PCR) assay[26]. When assessing 10 clinical Bundibugyo virus and 15 clinical EBOV Makona samples, concordant real-time PCR and MIPS positive results ranged from 9 out of 10 clinical samples (Table 1) to 6 out of 15 (Table 2), respectively. Intuitively, lower quantitation cycle ($C_q$) values correlated with higher MIPS read classification, suggesting the capability of ID-NGS to detect organisms was dependent on the starting concentration of the target genomic material. MIPS false negative calls for low target analytes

**Table 3 Experimental design and results from EBOV mock clinical trial**

| PFU/ml (LOD) | n | Avg EBOV reads | Avg %reads mapped | CoV | Positive samples | Negative samples |
|---|---|---|---|---|---|---|
| 1,000,000 (10×) | 16 | 5442.5 | 2.66% | 136.55% | 15 | 1 |
| 500,000 (5×) | 16 | 2777.5 | 2.49% | 152.33% | 13 | 3 |
| 100,000 (1×) | 16 | 351.5 | 0.58% | 247.57% | 9 | 7 |
| NTC | 100 | 4 | 0.00% | 571.69% | 1 | 99 |

Study design and demonstration of the preliminary diagnostic performance of an EBOV MIPS diagnostic assay. This table shows results from a mock clinical trial using 48 Zaire ebolavirus Makona positive samples at three different concentrations (10 ×, 5×, and 1 × ) and 100 Ebola negative samples

**Table 4 EBOV mock clinical trial diagnostic performance**

| N | Positive predictive value | Negative predictive value | Sensitivity | Specificity | Prevalence |
|---|---|---|---|---|---|
| 148 | 97.37% (83.95–99.62%) | 90.00% (84.26–93.80%) | 77.08% (62.69–87.97%) | 99.00% (94.55–99.97%) | 32.43% (24.98–40.61%) |

Study design and demonstration of the preliminary diagnostic performance of an EBOV MIPS diagnostic assay. This table shows the diagnostic performance of the EBOV MIPS mock clinical trial. Numbers in parentheses represent the 95% confidence interval

suggested that complete in silico validation is an unrealistic approach for clinical trials without comparison with some gold standard reference method, in this case real-time PCR.

Consistent concordance between the benchmark RT-PCR assay, the MIPS test device and the FDA-ARGOS in silico target sequence validation was important for establishing confidence in considering the in silico comparison method for clinical sample ID calling. To test this assumption, we used three read classification tools, MegaBLAST[24], Kraken[25], and LMAT[27] to evaluate the proposed in silico target sequence validation method (Fig. 1a), and to verify the potential for using in silico comparison without any empirical validation. MegaBLAST and Kraken analyses of raw sequence data for Bundibugyo virus samples in combination with FDA-ARGOS as the reference genome database showed complete agreement for MIPS and in silico calls (Table 1). Because the in silico comparison missed the classification call against the gold standard PCR benchmark test for a sample with low analyte levels (1 false negative result for the in silico validation), we performed a more in depth analysis of the additional EBOV Makona samples across MegaBLAST, Kraken and LMAT (Table 2). These analyses showed similar results to the Bundibugyo virus data at 100% agreement with the test device, but only for samples with low $C_q$ or high-input concentrations of the target organism. Additional analyses comparing results for each bioinformatics tool reference databases, with and without FDA-ARGOS genomes added, produced similar results demonstrating that FDA-ARGOS quality-controlled genomes alone were sufficient for in silico comparison (Supplementary Data 9). Overall, these data suggested in silico sequence comparison would be completely reliant on the inherent sensitivity of the sequencing assay to generate sequence read data, therefore the composite reference method (C-RM) (combining in silico target sequence comparison with representative clinical testing) is necessary for full validation of the test ID-NGS device. Figure 1a illustrates the C-RM, highlighting the need for empiric assessment of an ID-NGS assay-specific subset of samples or well-defined microbial reference materials (MRM).

Evaluation of the clinical samples suggested a need for benchmarking ID-NGS assays to currently implemented reference methods, thus the application of the C-RM. To document the application of MIPS Ebola Makona ID-NGS assay benchmarking, we performed a mock clinical trial to assess the representative clinical testing evaluation as part of the proposed

**Table 5 EBOV mock clinical trial prior probabilities**

| Prior probability of infection | Positive predictive value | Negative predictive value |
|---|---|---|
| 0 | 0 | 1 |
| 0.01 | 0.44 | 1 |
| 0.05 | 0.8 | 0.99 |
| 0.1 | 0.9 | 0.97 |
| 0.15 | 0.93 | 0.96 |
| 0.2 | 0.95 | 0.95 |
| 0.25 | 0.96 | 0.93 |
| 0.3 | 0.97 | 0.91 |
| 0.4 | 0.98 | 0.87 |
| 0.5 | 0.99 | 0.81 |
| 0.6 | 0.99 | 0.74 |
| 0.7 | 0.99 | 0.65 |
| 0.75 | 1 | 0.59 |
| 0.8 | 1 | 0.52 |
| 0.85 | 1 | 0.43 |
| 0.9 | 1 | 0.32 |
| 0.95 | 1 | 0.18 |
| 0.99 | 1 | 0.04 |
| 1 | 1 | 0 |

Study design and demonstration of the preliminary diagnostic performance of an EBOV MIPS diagnostic assay. This table shows positive and negative predictive values for prior probabilities of infection ranging from 0 to 1

C-RM. Initially, we performed a preliminary limit of detection (LOD) evaluation to determine the scope of the mock clinical evaluation. These experiments showed a preliminary LOD of $10^5$ with linear dose response correlation to EBOV input across the titration (Supplementary Table 3). An additional 40 positive replicates performed on two independent runs confirmed the LOD at $10^5$ pfu/ml for EBOV. This concentration formed the basis for spike-in levels of the mock clinical trial. From a total of 148 samples tested, 48 constituted real-time PCR positive spiked samples with 16 at high (10× LOD), 16 at medium (5× LOD), and 16 at 1× LOD for the MIPS assay (Table 3). Only 9 out of 16 samples at 1× LOD for the MIPS assay were positive with 37 out of 48 samples positive across the entire sample set in this analysis. However, the positive predictive value (PPV) and negative predictive value (NPV) for the MIPS assay were 97.4% and 90%, respectively, at or above the limit of detection with a

prevalence of 32.4% (Table 4). In addition, Table 5 lists the positive and negative predictive values for prior probabilities of infection from 0 to 1. The PPV and NPV metrics are important predictive analytics tools to provide performance characteristics for how the ID-NGS diagnostic test will perform in a clinical context. These data provide a rationale for developers using partial in silico validation when the false negative rate is low.

For future benchmarking efforts of bioinformatics tools, we have provided all Ebola reference datasets and databases (see Data availability section: Reference datasets c–e and Supplementary Data 1).

## Discussion

To encourage innovation in the infectious disease community, we provide our rationale and quality metrics for the FDA-ARGOS reference genome database and demonstrate the utility of the FDA-ARGOS database on the example of two use cases.

A critical aspect for assessing performance of any diagnostic is the availability of minimum quality control metrics for data, genomic or otherwise, for validation. Defined here are the FDA-ARGOS regulatory-grade genome criteria that provide ID-NGS diagnostic assay developers and the scientific community with traceable and quality-controlled reference genomes for diagnostic use. These high-quality genomes coupled with a streamlined approach for comprehensive expansion of FDA-ARGOS beyond the initial 2000 genomes is essential for continued ID-NGS diagnostic assay development.

FDA-ARGOS genome sequencing and research resulted in six broad quality metrics (Fig. 1b) defining regulatory-grade genome criteria required for current and future FDA-ARGOS contributors. All extant genomes in the FDA-ARGOS database (Supplementary Data 1) adhere to the quality metric of 95% coverage with 20× depth at every position across the entire assembled genome. This metric applies to the initial deposition of, minimally, five genomes of any genus/species added to FDA-ARGOS. These five genomes define the FDA-ARGOS core genome. After five or more regulatory-grade genomes per genus/species are available in the database, we will consider lower threshold metrics for FDA-ARGOS inclusion to capture unique genomes that may be diagnostically informative.

Idealistically, a reference genome model would provide 100% genome coverage with infinite depth encompassing all potential information content on the genomic level. This ideal would also capture additional layers of information such as genome plasticity (e.g., all genome variations) and similarity to all near-neighbors. This is unattainable with current technologies and without infinite comprehensive sampling of the organisms in question; however, FDA-ARGOS quality-controlled reference genomes provide a step in this direction. By accounting for human-made and technology errors and surveying species-specific and near neighbor genome variations, FDA-ARGOS begins the process of providing quality-controlled reference genomes for diagnostic relevant microbes. Currently, we believe thresholds at 95% coverage and 20× depth are sufficient to apply these genomes for diagnostic purposes within bounded use cases. An additional metric of minimally five representative genomes attempts to capture an initial understanding of genetic variation within an organism rather than relying on a specific type strain consensus sequence. Understandably, five representative genomes are not sufficient for many organisms to fully characterize the pan-genome and understand all genetic variations within a species. Hypothetically, a species-specific number of genomes exist that could determine encompassing genomic information about a microbe. Future efforts that focus on species-specific databases will be required to determine the species defining number of

genomes to robustly capture species diversity. As such, these requirements may change in the future as organisms are discovered, and technologies emerge.

We are working on methods to incorporate additional high-quality reference genomes through a qualification process utilizing FDA-ARGOS reference genome characteristics. More specifically, we are assessing genome quality (e.g., coverage, ANI, GC content, assembly size), genome continuity (e.g., N50, L50, number of contigs), taxonomy and metadata (e.g., species name, isolation source, submitter, orthogonal reference method) metrics to allow the community to qualify genomes for ARGOS deposition. These efforts will apply the above listed metrics to existing genomic information in the public domain coupled with machine learning methods and artificial intelligence to inform an external genome qualification tool greatly expanding the utility of the FDA-ARGOS database.

Lack of quality-controlled reference genomes challenges the accuracy of reference-based ID-NGS alignment for queryable microbial pathogens. Use case 1 highlights current challenges with infectious disease NGS technology when using minimal-, non-curated or absent reference databases potentially resulting in the lack of a diagnostic call or even misdiagnosis. These data were punctuated by two key findings: (1) de novo assembly of sequence data was not possible due to the low number of reads in clinical matrix and (2) varying quality of microbial reference genomes in publicly available databases made the metagenomic sample identification almost impossible (Supplementary Data 3, 4, 5). The latter point is extremely relevant for the intent of ID-NGS for diagnostic applications. In the case presented here, the top microbial sequence hit did not equate to the microbe of interest due to lack of quality control in the NCBI Nt reference database. Intuitively, use of quality-controlled FDA-ARGOS genomes mitigated this issue. Some level of contamination is expected, however, the simulation results with the normalized NCBI Nt databases suggest a necessity for more rigorous quality control, including human and lab contaminant screening of microbial reference databases, for reference-based ID-NGS alignment applications, especially for diagnostic use. In addition, *E. avium* isolate sequencing results showed the dependency of both classification algorithm (such as MegaBLAST and Kraken) and database used (Supplementary Data 6, 7, 8). This last aspect of the *E. avium* use case informed the consideration of the C-RM and opened the possibility for utilizing a suite of validated bioinformatics tools for in silico target sequence validation. In addition, these results clearly demonstrated the need for comprehensive citizen-science benchmarking studies to investigate the impact of different classification algorithms and criteria. The PrecisionFDA CDRH Biothreat Challenge addresses this need. More information on the challenge is available here (https://precision.fda.gov/challenges/3) and on the expert blog (https://precision.fda.gov/experts/6/blog).

There are two basic contrasting philosophies in circulation regarding genomic information and ID-NGS: (1) all information, whatever the quality, is useful towards making a diagnosis, the more data the better, with the assumption of diagnosis relying on error correction through iteration, or (2) quality-controlled, highly curated genomes are required as a solid foundation, more information is better, however, diagnostics require quality-controlled genomes to inform the basis of diagnosis. Experiments and data presented here support the latter of these two arguments. Specifically, while *E. avium* reference genomes were available in the normalized NCBI Nt database instances, positive ID-NGS identification of *E. avium* in the metagenomic sample required quality-controlled FDA-ARGOS reference genomes.

Over 2% of simulations with MegaBLAST and NCBI Nt revealed a mislabeled *E. hirae* as top hit for the *E. avium* isolate

data potentially leading to false positive species identification calls if this database was used. A potential mitigation for this issue would be the application of similar quality control metrics that are utilized for FDA-ARGOS reference genome inclusion. We fully support the use of NCBI Nt as a reference database, but with appropriate controls. The establishment of FDA-ARGOS within NCBI provides an additional resource specifically tailored for diagnostic purposes. In addition to addressing some of these controls, our primary goal for FDA-ARGOS reference genomes is to provide a tool to enable in silico validation efforts and advance innovation. NCBI Nt can be used in this context; however, with the understanding there may be an impact on performance of the downstream diagnostic.

Quality and coverage of targeted organisms are critical aspects for ID-NGS transition into the clinical space; however, to foster the transition, methods are required to lessen the burden for validating ID-NGS against all queryable pathogens. This manuscript documents methods for use of quality-controlled FDA-ARGOS reference genomes in in silico target sequence comparison as part of the proposed C-RM. We showed here that the in silico validation of Bundibugyo and Zaire ebolavirus can use FDA-ARGOS genomes as the comparator. For MIPS positive samples, there was 100% concordance between the gold standard real-time PCR comparator, and the in silico target sequence comparison. This supports the feasibility of implementing this strategy to shorten future clinical NGS-based assay evaluation studies. However, real-time PCR was more sensitive than the MIPS NGS assay especially at high $C_q$ values. A potential mitigation for this issue is the application of additional enrichment strategies to bring ID-NGS to similar sensitivities as the gold standard[28,29]. However, in the current form, observed lower sensitivity of the MIPS assay compared with real-time PCR shows the necessity for a C-RM and incorporating additional empirical studies, i.e., an assay-specific subset of clinical samples going through wet-lab comparison as part of the clinical validation. Discordant results at high $C_q$ values highlight the perils of solely applying in silico target sequence comparison. Without any empirical evaluation, in silico comparison would only provide results within the sensitivity ranges of the test ID-NGS device without providing the needed benchmark for sensitivity compared with a gold standard, such as real-time PCR. Therefore, as part of the C-RM, we demonstrate a preliminary performance assessment against a gold standard for a subset of the clinical trial samples with the intent that the remainder of the clinical trial samples could be validated via in silico sequence comparison. Different sample read depths may be required to achieve the desired identification performance for various organisms. Assay developers might be required to use an external comparator only for in silico validation results where the test device and in silico comparison yielded a discordant result. Potential limitations of the C-RM approach may be (1) cases where orthogonal reference method testing is not feasible, or (2) suitable reference genomes are not available. Mitigation strategies include potential use of other valid scientific evidence for orthogonal testing and submission of reference genomes to FDA-ARGOS for qualification and inclusion. We envision this C-RM to be a primary utility of the FDA-ARGOS genome database tool for medical device development. We hope that FDA-ARGOS will spur innovation and expedite regulatory science, and ultimately enable ID-NGS as a diagnostic to enter the clinic.

The FDA-ARGOS reference genome resource is a constantly evolving public database intended to mature over time with community support and genomic technology advancements. Continued population and expansion of the FDA-ARGOS database resource will be required to cover the panoply of infectious microorganisms. The need for comprehensive regulatory-grade genome coverage is clear, however, no one entity can perform all the needed sequencing. We are therefore working on a pathway for external genome qualification to streamline and expand FDA-ARGOS resource as needed. Both the external genome qualification and continued research to apply this regulatory-grade standard to unculturable and emerging pathogens will be the focus of future research.

Further population and curation of the database will support the success of FDA-ARGOS and promote adoption by the NGS community. The FDA team is looking for unique, hard-to-source microbes like biothreat organisms, emerging pathogens and AMR-related pathogens to help improve the database. We encourage the community to share microbe samples here (https://argos.igs.umaryland.edu/).

## Methods

**FDA-ARGOS database genome deposition**. Using previously identified microbe (s), nucleic acid was extracted for library preparation and sequencing. Next, microbial nucleic acids are sequenced, and de novo assembled using Illumina and Pac Bio sequencing platforms at the Institute for Genome Sciences at the University of Maryland (UMD-IGS). The assembled genomes were quality controlled by an ID-NGS subject matter expert working group consisting of FDA personnel and collaborators with all passing data deposited in NCBI databases. Follow this link (https://www.fda.gov/medical-devices/science-and-research-medical-devices/database-reference-grade-microbial-sequences-fda-argos) for full background, collaborators and FDA-ARGOS genome status. Supplementary Data 1 lists all FDA-ARGOS genomes with accessions and statistics used in this manuscript.

**Bacterial reference genome sequencing and assembly**. A hybrid sequencing approach[22] based on long and short read NGS technology was selected using Illumina and PacBio NGS technologies to generate high-quality bacterial genome sequences. Sufficient and high molecular weight genomic starting material was needed for both technologies. Sets of bacterial libraries were multiplexed on the Illumina PE HiSeq4000 using the 150 bp paired-end run protocol with 24–48 isolates per lane. The coverage threshold was set at 300× to ensure sufficient read depth was achieved from short read NGS technology for high-quality assembly generation. In addition, sets of bacterial libraries were run on the PacBio RS II P6-C4 with at least one SMRT cell per bacterial genome. The coverage threshold was set at 100× to ensure sufficient and economically feasible read depth was achieved from long read NGS technology for high-quality assembly generation. The data were assembled both separately and in combination using a series of assembly tools, including SPAdes[30], Canu[31], HGAP[32], and Celera Assembler[33]. Pilon[34] was used for polishing of data. Manual curation was performed to achieve optimal assembly and consensus calling.

**Viral reference genome sequencing and assembly**. Viral genome sequencing included shotgun, amplicon, and optional 5′/3′ RACE sequencing methods to generate full-length viral genome sequences. Sufficient and high-quality genomic starting material was needed for all three approaches. Amplicon sequencing with 48–96 overlapping amplicons was used to generate deep coverage of known regions of the genome and was used to evaluate quasi-species in each isolate. Rapid amplification of cDNA Ends (RACE) was used when desirable to finish the 5′ and 3′ ends, and a shotgun approach generated data from all RNAs present in the sample without the level of bias present in the amplicon approach. Sets of viral libraries from all three approaches were multiplexed on the Illumina MiSeq using the 300 bp paired-end run protocol. The coverage threshold was set at 100× to ensure two times amplicon coverage across the genome. The shotgun, amplicon and RACE data were assembled both separately and in combination using a series of assembly tools, including SPAdes[30] and Celera Assembler[33]. Manual curation was performed to achieve optimal assembly and consensus calling.

**FDA-ARGOS genome assembly quality control statistics**. Coverage statistics were calculated for each of the FDA-ARGOS genome assemblies. Illumina coverage and PacBio coverage were calculated separately. Illumina short reads were first aligned to the assembly consensus sequence using Bowtie2[35]. Illumina coverage was then calculated using samtools[36] on the resulting sam file. PacBio reads were aligned to the assembly consensus sequence using BLASR[37]. PacBio coverage was then calculated using samtools[36] on the resulting sam file. Total coverage was calculated by adding the PacBio coverage and Illumina coverage at every base pair location in the assembly consensus sequence.

**FDA-ARGOS genome annotations**. Genomes were annotated with NCBI's annotation tools to streamline the process[38–42]. Bacterial sequences were annotated with NCBI's Prokaryotic Genome Annotation Pipeline (PGAP) that combines *ab initio* gene prediction algorithms with homology-based methods. Viral sequences were aligned with their most similar NCBI RefSeqs (NC_002549, NC_014372, NC_006432, NC_014373, NC_004162, NC_004161, NC_003899, NC_001449, NC_001544, NC_035889), using the Geneious alignment tool in the Geneious platform[43]. The setting to automatically determine detection was used, and the other parameters were set to the defaults. Gene, CDS, and mature peptide annotations from the RefSeqs were transferred to the sequences, beginning and end positions were verified for homology, and the sequences were manually reviewed for unexpected stop codons or regions of high dissimilarity. The RefSeqs used have had their annotation reviewed by NCBI curators based on available literature, and in several cases, the annotations were performed in collaboration with researchers familiar with the viruses.

**Clinical sample collection and preparation**. Clinical and mock clinical sample testing was conducted to demonstrate the utility of FDA-ARGOS. Ten de-identified human serum samples that were suspected Bundibugyo virus positive were received from the Democratic Republic of Congo (DRC). These samples were determined by the USAMRIID Office of Human Use and Ethics to be Not Human Subject Research (HP-12–15). Presence of virus for the human samples was determined using the previously established Bundibugyo virus real-time RT-PCR assay[26]. Samples were run in duplicate using 5 µl of purified RNA on the Light-Cycler 480 (Roche Diagnostics Corporation). A positive sample was defined as having a quantitation cycle ($C_q$) value of < 40 cycles (Table 1).

Fifteen de-identified human serum samples that were Ebola virus (EBOV) Makona positive were received from Sierra Leone; these samples were determined by the USAMRIID Office of Human Use and Ethics to be Not Human Subject Research (HP-09-32). All samples were collected and de-identified in Sierra Leone at the Kenema Government Hospital, and the samples had indirect identifiers upon receipt. Presence of virus for the human samples was determined using the previously established real-time RT-PCR assay[26]. Samples were run in duplicate using 5 µl of purified RNA on the LightCycler 480 (Roche Diagnostics Corporation). A positive sample was defined as having a quantitation cycle ($C_q$) value of < 40 cycles with duplicate positive real-time PCR results (Table 2).

One clinical *E. avium* from Children's Hospital was used for this study and maintained at USAMRIID through the Unified Culture Collection (UCC) system. Following overnight growth of *E. avium* (~16 h), a single, isolated colony was chosen and inoculated into tryptic soy broth (ThermoFisher, Waltham, MA). A glycerol stock was made from the overnight culture and colony counts were performed concurrently to determine the CFU/mL of the stock organism.

**Metagenomic and isolate shotgun sequencing**. The *E. avium* sample SAMN04327393 was cultured on blood agar plates or in tryptic soy broth (ThermoFisher, Waltham, MA). Samples were spiked to a final concentration of $10^5$ CFU/ml in water or whole blood matrix (BioreclamationIVT, Baltimore, MD) and 100 µl was extracted using the Qiagen EZ1 viral kit (Qiagen, Valencia, CA) according to the manufacturer's instructions. DNA concentration was quantified utilizing Qubit dsDNA BR assay kit (ThermoFisher). DNA samples were prepared for sequencing on the MiSeq platform utilizing the Nextera XT DNA library preparation kit according to the manufacturer's instructions (Illumina, San Diego, CA). Library preparations were quantified and normalized utilizing the KAPA library quantification kit (Kapa Biosystems, Wilmington, MA) and sequenced on the MiSeq platform using the 2 × 150 cycle sequencing kit (Illumina). Sequencing reads were analyzed using CLC Genomic Workbench (CLC Bio, Cambridge, MA). For metagenomic analysis, paired-end reads were trimmed on CLC using a modified-Mott trimming algorithm utilizing a quality trim of 0.05 and reads below 50 bp in length were removed from further analysis. Quality scores in CLC are on a Phred scale and Phred quality scores ($Q$) are defined as $Q = -10\log10(P)$ where $P$ is base calling error probability. Trimmed reads were then mapped to *E. avium* assembly GCF_000407245.1 and *H. sapiens* assembly GCA_000001405.27 using CLC genomics workbench v 10.1.1. Mapping parameters were as follows: mismatch costs = 2, insertions costs = 3, deletion costs = 3, length and similarity fraction = 0.8.

**Targeted molecular inversion probe sequencing (MIPS)**. The Bundibugyo virus (BDBV) and Ebola virus (EBOV) Makona clinical data samples were run using the MIPS approach[44] to capture a targeted sequence into a circular oligonucleotide. A PCR reaction and subsequent NGS on the Illumina MiSeq (2 × 150) amplified and identified the captured sequence using CLC genomics workbench (CLC Bio, Cambridge, MA) read mapping back to the reference genome (EBOV (GenBank # NC_002549), BDBV (GenBank # NC_014373). A positive call was determined for each sample as any reference which had total sequencing reads above a cutoff value as determined by the average plus three times the standard deviation of three independently run non-template control samples. The percent reads classified as

Bundibugyo virus or EBOV Makona was reported. For the MIPS approach, the remaining reads are non-specific or background.

**Mock clinical diagnostic evaluation**. The MIPS assay was evaluated for diagnostic performance across 148 blinded samples. The limit of detection (LOD) was determined through a preliminary titration of EBOV Zaire in TRIzol starting at $10^8$ plaque forming units (pfu)/ml down to $10^2$ pfus/ml and then run in triplicate. The concentration where all three replicates yielded positive results was confirmed as the LOD across 40 replicates at that concentration. EBOV (Kikwit R4317a) in TRIzol LS was diluted to $10 \times (1.0E + 06$ pfu/ml), $5 \times$ ($5.0E + 05$ pfu/ml), and $1 \times (1.0E + 05$ pfu/ml) LOD in triplicate in matrix also containing TRIzol LS. Nucleic acid was extracted using 400 µl of each sample, along with 14 negative serum samples, on the EZ1 Virus 2.0 kit and eluted in 60 µl. Presence of virus was determined with an established real-time PCR assay in triplicate for each extracted sample. Extracted RNA was amplified from 5 µl total nucleic acid using the Quantitect Whole Transcriptome Amplification Kit (Qiagen) and quantified with the Qubit dsDNA Broad Range Assay Kit. A total of 50 ng cDNA was added into the MIP protocol. Library preparation was performed on the Apollo instrument using the PrepX Complete ILMN 32i DNA kit and Illumina TruSeq dual Indices. For the mock clinical evaluation, 48 positive and 100 negative (matrix only) samples were sequenced on the Illumina MiSeq using the 300 cycle kit. Sixteen positive samples were spiked at $10\times$, $5\times$, and $1\times$ LOD. Threshold cutoffs for positive samples were $2\times$ signal to noise ratio (SNR). All diagnostic performance statistics were calculated on https://www.medcalc.org/calc/diagnostic_test.php.

**Short read processing and database construction**. The quality of the short reads was checked with FastQC. No quality trimming was conducted. We selected 1,000,000 short reads randomly from each of the samples. 200 database instances were generated by randomly sub-sampling NCBI GenBank assemblies and the FDA-ARGOS assemblies (Supplementary Data 2, SAMN04327393 was excluded from the reference database construction for the case studies because this genome was developed from the same isolate that was used as spike-in material for use case 1). Each database instance has the same species composition and identical number of assemblies per species. The exact species composition of the assembly sets was determined by finding an intersection of the FDA-ARGOS assemblies and GenBank assemblies.

**Short read classification**. The MegaBLAST function of blast + 2.7.1 installed on FDA HPC infrastructure (https://www.ncbi.nlm.nih.gov/books/NBK153387/) was used to taxonomically classify the short reads using the default parameters and the 200 database instance assembly sets. Each of the 200 database instance assembly sets was made into a nucleotide database using the makeblastdb command. For this study, the taxon associated with the first reported alignment sorted by max alignment score was used as the taxonomic label for each read. Original Mega-BLAST results were summarized to report the number of reads associated with each unique NCBI taxonomy ID called.

Kraken 1.0, installed on FDA HPC infrastructure (https://ccb.jhu.edu/software/kraken/MANUAL.html), was used to assign a taxonomic label to each short read using default parameters and the same 200 database instance assembly sets. The database instance assembly sets were built into Kraken databases using the default options. Original Kraken results were summarized to report the number of reads associated with each unique NCBI taxonomy ID called.

LMAT version 1.2.6 (available for download at sourceforge.net/lmat), installed on Lawrence Livermore National Laboratory (LLNL) HPC infrastructure was used to assign a taxonomic label to each short read with a minimum score setting of 0.5. Match scores are calculated per read, by fitting a random null model created by simulating 1 GB of random sequence for each model dependent on read length and GC content. Three databases, the Algorithm Standard Database (LMAT DB), the stand-alone FDA-ARGOS, and an aggregated database consisting of both the LMAT DB database and the stand-alone FDA-ARGOS database were used. LMAT results were summarized to report the number of reads associated with each unique NCBI taxonomy ID.

**Reporting summary**. Further information on research design is available in the Nature Research Reporting Summary linked to this article.

## Data availability
All FDA-ARGOS reference genome raw data, assemblies, annotations, metadata, base modification data and pipeline information are available from Bioproject ID# PRJNA231221 and at https://www.ncbi.nlm.nih.gov/bioproject/231221. The five reference datasets from the use cases are available from Bioproject ID# PRJNA495928 and at https://www.ncbi.nlm.nih.gov/bioproject/495928:
- Metagenomic shotgun sequencing for identification of *E. avium* (three replicate samples).
- Isolate shotgun sequencing for identification of *E. avium* (three replicate samples).
- MIPS for identification of Bundibugyo virus (10 PCR positive, 1 NTC).
- MIPS for identification of Ebola virus Makona (15 PCR positive, 1 NTC).

- MIPS EBOV mock clinical trial (148 blinded samples: 48 positive: 16 10 × LOD, 5 × LOD, 1× LOD, and 100 matrix-only negative).

All other data are contained within this article and its supplementary information or are available from the corresponding author upon reasonable request.

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

## Acknowledgements

This project has been funded with Federal funds from the Office of Counterterrorism and Emerging Threats, Food and Drug Administration, Department of Health and Human Services, under Contract No. HHSF223201310109C, HHSF223201510106C, HHSF223201610073C and the Department of Defense, under Contract No. 224-15-6506R. This research was supported by the Intramural Research Program of the NIH, National Library of Medicine. This project was funded by DTRA, Contract No. CB10245. The authors would like to thank Dr. Sally Hojvat and Dr. Peyton Hobson for helpful discussions. Sample contributions for the initial set of 500 from the U.S. Army Medical Research Institute of Infectious Diseases, the Department of Defense Critical Reagents Program, Public Health Agency Canada, Public Health England, the University of Texas Medical Branch, BC Centre for Disease Control, American Type Culture Collection, Rockefeller University, FDA-CBER (Maria Rios, Robert Duncan, Rafaelle Gusmao), FDA-CFSAN (Eric Brown, Marc Allard, Maria Hoffman, Cary Pirone), FDA-CVM (Patrick McDermott, Shaohua Zhao), Children's National Hospital (Joseph Campos, Brittany Goldberg, Chelsie Geyer), University of Colorado School of Medicine (Thomas Morrison, Sudhakar Agnihothram). The opinions, interpretations, conclusions, and recommendations contained herein are those of the authors and are not necessarily endorsed by the U.S. Army. The views expressed here are those of the authors and do not necessarily represent the views or official position of the FDA, NIH, or DOE.

## Author contributions

H.S. conceived of the project, led the project, collected samples, registered samples, wrote and revised the paper, generated figures, tables and data, performed data analysis, and served as the principal investigator. T.D.M. led the coordination of the use cases and wrote and revised the paper. Y.Y. did script/command development, data analysis, gathered and organized FDA-ARGOS database metrics. C.S collected and isolated

samples, extracted DNA, performed library preparations and Illumina sequencing, gathered and organized data for the *E. avium* gap study, and generated and revised figures. A.H. performed DNA extraction, library preparation, MIPS and Illumina sequencing, and gathered and organized data for the Ebola in silico study. L.T. did IGS sequencing work. L.S. did IGS sequencing work. S.N. did data analysis, and gathered, registered in NCBI and organized data from IGS sequencing work. W.K. and M.S. helped with coordination of BioProject and data submissions and bacterial annotations and the assessment for gap filling. E.H. did viral annotations. D.L. did LMAT analysis. J.A. coordinated LMAT analysis. J.K. collected and sequenced samples. T.S. helped develop the study and experimental design. S.L. helped develop the study and experimental design. R.S. collected clinical samples. U.S. helped develop the study and experimental design.

## Additional information

**Competing interests:** The authors declare no competing interests.

