## [Peer Review File · Nature Communications]

Reviewers' Comments:

Reviewer #1:

Remarks to the Author:

In this manuscript, Sichtig and colleagues describe the development and benchmarking of the publicly available FDA-ARGOS database, designed to be a regulatory-grade database that addresses existing database gaps with quality-controlled genomic sequences. In addition to presenting quality control metrics, they also describe 3 case studies highlighting potential applications of FDA-ARGOS for infectious disease diagnostics: (1) as an in silico comparator tool, (2) to identify *Enterococcus avium* in a contrived cultured genome spiked into matrix, and (3) to identify Ebola virus in clinical serum samples. The authors are to be commended for the development of what will likely be a valuable public database resource for researchers, laboratory physicians, and clinical assay developers. We do need more comprehensive and accurate databases, and the existence of and ongoing additions to the FDA-ARGOS database are critical to addressing this need. The C-RM approach is novel and an elegant method to benchmark ID-NGS assays. Overall, the findings of this manuscript and the presentation of the FDA-ARGOS database are of great interest to the genomics community and the fields of microbiology, microbiome, biothreat surveillance, public health, and infectious diseases. The database is public and open to further submissions by others. However, I believe that the selective advantages of the database have been overstated and related to this have several concerns about the case studies and bioinformatics analyses as presented that should be addressed.

1. First, a major premise of the manuscript is that when considering alternatives of “the more data the better” versus the need for “quality-controlled, highly curated genomes”, the “experiments and data presented here support the latter of these two arguments”. I don't believe that the results as currently presented justify these claims. For example, in Figure 4, it's clear that the major factor in sensitivity of identification is the availability of genome sequences and not the quality of those sequences per se. It's obvious that Kraken did not identify *E. avium* because its database doesn't have any representation of *E. avium* genomes. When FDA-ARGOS *E. avium* genomes are added, either alone or in combination with the standard Kraken database, Kraken is able to detect *E. avium*. The authors should add existing *E. avium*, non-FDA-ARGOS sequences to the Kraken database and rerun the algorithm (as well as MegaBLAST) as a more fair and appropriate comparison. This would allow them to address the critical question of whether incompleteness of databases or quality of genome sequences is more important in determining the performance of reference-based ID-NGS alignment.

2. Also, for MegaBLAST, there were fewer reads identified specifically as *E. avium* when using the standard database in combination versus using the FDA-ARGOS alone. I would argue that this is a good thing – in contrast with the views of the authors (lines 469-472) -- because presumably the standard database (NCBI Nt) has more genomes of related organisms such as *E. faecium*, *E. faecalis*, *E. gallinarum*, etc. so that MegaBLAST is appropriately taxonomically classifying the reads so that only reads that are species-specific to *E. avium* are included. If this is the case, it is an example of the limitations of using a more limited, albeit higher-quality database such as FDA-ARGOS versus NCBI nt. The authors should comment on this.

3. I do not believe that FDA-ARGOS as an incomplete, albeit high-quality database would be superior to NCBI Nt, a variable quality, yet comprehensive database. Having fewer sequences limits your ability to appropriately taxonomically classify organisms; for instance, we have found that the more genome representation in your database, the better you are able to distinguish between closely related species such as *Shigella flexneri* and *Escherichia coli*. It is true that misannotations and bias in databases are an issue; however, the comprehensiveness alone would lead one to prefer end-users to adopt NCBI Nt rather than FDA-ARGOS at present (FDA-ARGOS is severely limited with respect to viruses and parasites, for instance), at least until FDA-ARGOS grows much larger. It seems to me that a combination of NCBI Nt and FDA-ARGOS would be at present the best database to use, instead of reliance on FDA-ARGOS alone. This should be

discussed in-depth in the manuscript.

4. A more direct comparison of FDA-ARGOS with “standard databases” would be if the authors took the curated, finished FDA-ARGOS genomes and simulated the incomplete and error-prone genomes that may exist in NCBI nt, and then used this modified FDA-ARGOS database for purposes of in silico comparison. This would be probably the best method to separate the effects of database size/completeness and database quality. It's unclear why the authors did not do so.

5. It would be helpful to the readers to understand the justification for why the authors chose their threshold metrics for the FDA-ARGOS database (i.e. 95% coverage at 20X depth, at least 5 representative genomes in that species) as constituting “regulatory-grade” genomes.

6. For the *E. avium* case, it would be useful to make clear in the figures about how many isolates of *E. avium* in NCBI Nt were remaining after removing SAMN04327393 versus how many isolates of *E. avium* were in FDA-ARGOS (n=4 is mentioned but only in line 486). The authors state that “..reads classified as *E. avium* ranged from an average 3829 and 840 when FDA-ARGOS genomes were added...compared to an average 29 and 0 reads when these genomes were absent...”. I find it hard to believe that there were only 29 reads identified using MegaBLAST (versus 3829 with FDA-ARGOS) if there were 4 genomes of *E. avium* in NCBI Nt, no matter how poor the quality. Furthermore, it is a little deceptive to state that *E. avium* reads were 840 for Kraken with FDA-ARGOS but 0 for Kraken because the standard Kraken database doesn't contain *E. avium* sequences. A more fair assessment, as mentioned previously, is to add the NCBI Nt *E. avium* sequences to the Kraken database and re-run the algorithm.

7. For the Ebola case study, why were only FDA-ARGOS EBOV genomes assessed? EBOV genomes are well represented in NCBI Nt especially due to the amount of sequencing done during the West African EBOV outbreak and the ongoing recent outbreaks in the DRC. Why are FDA-ARGOS EBOV sequences needed specifically? Why can't a representative subset of EBOV sequences from NCBI Nt be used for the C-RM approach? It is stated that “FDA-ARGOS alone was sufficient for in silico comparison”. However, wouldn't a random selection of representative EBOV sequences (non FDA-ARGOS) from NCBI Nt also be “sufficient”?

8. The C-RM approach is elegant but still requires a gold standard. This needs to be underscored in the discussion, as for many diseases, there may not be a gold standard clinical diagnostic test. A particularly relevant example is early acute Lyme disease (*Borrelia burgdorferi* PCR has low sensitivity; two-tiered antibody testing takes 2-4 weeks). This should be mentioned as a potential limitation of in silico validation using C-RM.

MINOR POINTS:

1. The authors mention using two different read technologies, PacBio and Illumina, in generating consensus sequences. Is this really needed? Arguably, the long reads may be needed for gene synteny but this would be primarily useful for unknown and/or novel reference genomes rather than genomes for which there is at least one high- or moderate-quality reference in NCBI nt. Furthermore, the use of even PacBio and Illumina may not even be sufficient to finish genomes with highly repetitive genome structures such as *Prevotella copri*, as elegantly described by Bhatt, et al and colleagues (https://www.nature.com/articles/nbt.4266?WT.feed_name=subjects_genome-informatics), in which nanopore sequencing and de novo assembly were required. This should be mentioned somewhere in the manuscript.

2. Regarding lines 360-365, it would be helpful for more description of what is meant by “gaps” in the database that FDA-ARGOS sought to address. Are these genomes? Near neighbors?

3. The authors should mention that finishing the ends of viral genomes with 5'/3' RACE, while critically important in virology such as development of an infectious clone, is likely less important for the purposes of reference-based NGS alignment, as the ends of viruses are usually not well-represented in metagenomic sequencing (especially with library preparation techniques using transposons such as Nextera which cleave internal to the sequence).

4. It might be preferable to describe the case studies as 2 rather than 3, as the first case study appears to be embedded with the EBOV example.

Reviewer #2:

Remarks to the Author:

In their manuscript "FDA-ARGOS: A Public Quality-Controlled Genome Database Resource for Infectious Disease Sequencing Diagnostics and Regulatory Science Research", Dr. Sichtig et al. describe extensive efforts to generate a publicly available database as a tool to support innovation of emerging technologies and provide an in-silico comparator tool that could reduce the burden for completing ID-NGS clinical trials. The manuscript also discusses quality control metrics for the proposed FDA-ARGOS database and presents three examples demonstrating potential applications for FDA-ARGOS in infectious disease diagnostics.

Major Comments

1) This reviewer congratulates the authors for undertaking a very important effort in generating a reference database of regulatory-grade, high-quality genome sequences. In several of the discussed applications of ID NGS, it will be common to generate sequencing data for microbes and pathogens that have no identical reference genome available (as the full genomic diversity of bacteria, viruses, and eukaryotic pathogens is not known and continuously changing). While likely not of higher quality, test-specific databases may be larger than the FDA-ARGOS database. It is therefore possible that test-specific databases will contain one or more reference sequences that are more similar to a given organism identified by a new test. In this case, analysis of the same raw sequencing data with the FDA-ARGOS database may produce different (but not necessarily wrong) results. The authors should consider discussing the limitations of inherently incomplete databases and potential pathways to resolving discrepancies that result from varying database completeness.

Clearly, scalability of efforts required to create curated and comprehensive databases is a concern. The authors briefly discuss how scalability of the FDA-ARGOS database could be increased (lines 589-592). Could the authors expand on the strategy and timeline for including existing genomic data into a regulatory-grade database? What quality control metrics could be implemented to include existing (rather than newly-generated) genome data into the FDA-ARGOS database?

2) Lines 368-369: "An initial collection criterion focused on sequencing at least 5 diverse isolates per species"

The authors may want to discuss how intra-species diversity may affect completeness of genetic representation by 5 isolates. This number may be too small for some taxa, especially bacteria with high genetic diversity and viruses.

3) Use Case 1

This contrived sample contained ~4000 sequencing reads of *E. avium*, representing approximately 10-15% of the bacterial genome. Do the authors consider this a sufficient representation for a positive result? This reviewer does not necessarily disagree with a positive interpretation. Given the intended use of the FDA ARGOS database, it would be valuable to the audience to learn more about how the authors think about defining true positive results.

Minor Comments

- Lines 297-298, 312-313: "For this study, the taxon associated with the first reported alignment was used as the taxonomic label for each read..."

Albeit easy to implement, this approach oversimplifies the complexity of the classification problem. For example, 'assigning the taxonomic label for each read to the first reported alignment' is highly dependent on the accuracy of reference sequence annotations, which are frequently incorrect or incomplete in the NCBI nt database. In addition, sequencing reads may map equally well to reference sequences with different taxonomic annotations. In this case, simply assigning the sequencing read to the 'first reported alignment' (presumably sorted by max alignment score).

- Lines 249-250: "...paired end reads were trimmed utilizing a quality of 0.05 and reads below 50bp in length were removed..."

Please provide additional details for quality trimming.

- Lines 250-253: "Trimmed reads were then mapped to *E. avium* assembly GCF_000407245.1 and *H. sapiens* assembly GCA_000001405.27. Mapping parameters were as follows: mismatch costs=2, insertions costs=3, deletion costs=3, length and similarity fraction = 0.8."

What mapping tool was used for this analysis?

- Line 262: "The threshold for positive calls was determined by the no template control (NTC)." This sentence is not entirely clear. Was the number of reads obtained with the NTC defined as the threshold for a positive call? If so, the use of an NTC for this purpose may provide a false sense of security as higher read counts may be obtained in samples positive for related viruses (causing false-positive results).

- Lines 360-362: "FDA, DOD, NCBI and other agencies using scientific literature, a phylogenetic data mining approach, and FDA microbial species-specific guidance documents identified more than 1000 gaps in public microbial genomic repositories. We prioritized these gaps and selected biothreat microorganisms, common clinical pathogens and closely related species (See Supplemental Materials for the organism gap list)."

Which supplemental file are the authors referring to?

- Lines 449-451 "For frame-of-reference, we would need over 30,000 reads to de novo assemble an entire genome of approximately 5 Mb at 1X coverage, assuming a read size of 150 bp and perfect quality of each generated read at all positions"

The authors may want to choose a more realistic example. De novo assembly of a bacterial genome is not possible with 1X coverage.

- Lines 465-467 "Interestingly, while *E. avium* genomes were available in the NCBI nt database and part of the read classification for MegaBLAST analyses, positive ID-NGS identification required the addition of quality-controlled FDA-ARGOS reference genomes"

Did the authors assess whether the *E. avium* genomes were available in the NCBI nt database were complete? Were they correctly identified? What was the average nucleotide identity of SAMN04327393 with the *E. avium* genomes contained in the NCBI nt database? Please provide additional information for the difference in results.

- Lines 488-489 "These top hits were potentially database contaminants and illustrate the risk of using non-curated databases in ID-NGS diagnostics"

In addition, the authors may want to consider discussing the potential impact of different classification algorithms or criteria.

- Lines 614-615 "Experiments and data presented here support the latter of these two arguments."

This reviewer does not disagree with this conclusion for the given examples. However, this was in part because the FDA ARGOS database included reference genomes for the studied examples. The

authors should discuss how results may differ in cases where a high-quality genome is not available for a pathogen contained in a given sample.

Reviewer's comments:

Note: We highlighted all edits in response to reviewer comments throughout the paper.

Reviewer #1 (Remarks to the Author):

In this manuscript, Sichtig and colleagues describe the development and benchmarking of the publicly available FDA-ARGOS database, designed to be a regulatory-grade database that addresses existing database gaps with quality-controlled genomic sequences. In addition to presenting quality control metrics, they also describe 3 case studies highlighting potential applications of FDA-ARGOS for infectious disease diagnostics: (1) as an in silico comparator tool, (2) to identify *Enterococcus avium* in a contrived cultured genome spiked into matrix, and (3) to identify Ebola virus in clinical serum samples. The authors are to be commended for the development of what will likely be a valuable public database resource for researchers, laboratory physicians, and clinical assay developers. We do need more comprehensive and accurate databases, and the existence of and ongoing additions to the FDA-ARGOS database are critical to addressing this need. The C-RM approach is novel and an elegant method to benchmark ID-NGS assays. Overall, the findings of this manuscript and the presentation of the FDA-ARGOS database are of great interest to the genomics community and the fields of microbiology, microbiome, biothreat surveillance, public health, and infectious diseases. The database is public and open to further submissions by others. However, I believe that the selective advantages of the database have been overstated and related to this have several concerns about the case studies and bioinformatics analyses as presented that should be addressed.

We thank the reviewer for the remarks and the opportunity to address concerns about the case studies and bioinformatics analyses.

1. First, a major premise of the manuscript is that when considering alternatives of “the more data the better” versus the need for “quality-controlled, highly curated genomes”, the “experiments and data presented here support the latter of these two arguments”. I don't believe that the results as currently presented justify these claims. For example, in Figure 4, it's clear that the major factor in sensitivity of identification is the availability of genome sequences and not the quality of those sequences per se. It's obvious that Kraken did not identify *E. avium* because its database doesn't have any representation of *E. avium* genomes. When FDA-ARGOS *E. avium* genomes are added, either alone or in combination with the standard Kraken database, Kraken is able to detect *E. avium*. The authors should add existing *E. avium*, non-FDA-ARGOS sequences to the Kraken database and rerun the algorithm (as well as MegaBLAST) as a more fair and appropriate comparison.

This would allow them to address the critical question of whether incompleteness of databases or quality of genome sequences is more important in determining the performance of reference-based ID-NGS alignment.

We thank the reviewer for the suggestions. We revised our use case 1 to address the impact of genome quality on performance of reference-based ID-NGS alignment applications.

More specifically, we generated 200 randomized database instances by sub sampling NCBI GenBank and FDA-ARGOS assemblies. Each database instance has the identical species composition and number of assemblies per species. The exact species composition of the assembly sets was determined by finding an intersection of the FDA-ARGOS assemblies and GenBank assemblies. We ran 2400 simulations using triplicate *E. avium* metagenomics and isolate samples in combination with the normalized NCBI and FDA-ARGOS database instances. A summary and the underlying raw data are available in updated Supplemental Tables 4, 4a, 4b, 5, 5a and 5b. We updated Figure 4 with a heatmap visualization of the MegaBLAST tool results from triplicate metagenomics *E. avium* samples.

We believe that these new simulations provide a more fair comparison. MegaBlast and Kraken tools were used for bioinformatics analyses. Our updated results clearly demonstrated that genome quality critically impacts final call performance.

2. Also, for MegaBLAST, there were fewer reads identified specifically as *E. avium* when using the standard database in combination versus using the FDA-ARGOS alone. I would argue that this is a good thing – in contrast with the views of the authors (lines 469-472) -- because presumably the standard database (NCBI Nt) has more genomes of related organisms such as *E. faecium*, *E. faecalis*, *E. gallinarum*, etc. so that MegaBLAST is appropriately taxonomically classifying the reads so that only reads that are species-specific to *E. avium* are included. If this is the case, it is an example of the limitations of using a more limited, albeit higher-quality database such as FDA-ARGOS versus NCBI nt. The authors should comment on this.

While NCBI Nt has more genome diversity our updated use case 1 clearly demonstrated that quality matters and is expected to impact more accurate species-specific organism identification.

For use case 1, we sequenced an *E. avium* metagenome (*E. avium* at 10^5 in mock clinical matrix) and isolate sample. Our expectation was to see low level *E. avium* reads in the metagenome sample. The species identification algorithms (MegaBLAST, Kraken) were connected to 200 normalized NCBI Nt database (varying quality) and FDA-ARGOS database instances and contained the same number of randomly chosen *E. avium*, *E. faecium*, *E. faecalis*, *E. durans*, *E. gallinarum* and *E. hirae* reference genomes. Our results from 2400 simulations showed that high quality reference genomes enable more precise species-specific calls.

Most outstanding were the 1200 metagenome simulations. *E. avium* was called consistently at an average read number of 61 with the FDA-ARGOS and normalized NCBI Nt database instances. However, an average 20% of reads mapped to other microbial species with the normalized NCBI Nt database instances. Here, *Vibrio vulnificus* and *E. faecium* were called at over 10% each in the metagenome sample (Supplemental Table 4, 4a, 4b). This presents a dilemma for species identification applications where sample makeup is presumably unknown. In our cases, 33 (MegaBLAST) and 39 (Kraken) species would need to be ruled out before the correct call could be made. Many of these species calls are clinically relevant and well above a 1% mapped reads cutoff.

The 1200 isolate simulations revealed that *Enterococcus* genus was consistently called with *Enterococcus avium* as the top species. To our surprise, several simulations with the normalized NCBI Nt database showed *Enterococcus hirae* as top hit (Supplemental Table 5, 5a, 5b).

3. I do not believe that FDA-ARGOS as an incomplete, albeit high-quality database would be superior to NCBI Nt, a variable quality, yet comprehensive database. Having fewer sequences limits your ability to appropriately taxonomically classify organisms; for instance, we have found that the more genome representation in your database, the better you are able to distinguish between closely related species such as *Shigella flexneri* and *Escherichia coli*. It is true that misannotations and bias in databases are an issue; however, the comprehensiveness alone would lead one to prefer end-users to adopt NCBI Nt rather than FDA-ARGOS at present (FDA-ARGOS is severely limited with respect to viruses and parasites, for instance), at least until FDA-ARGOS grows much larger. It seems to me that a combination of NCBI Nt and FDA-ARGOS would be at present the best database to use, instead of reliance on FDA-ARGOS alone. This should be discussed in-depth in the manuscript.

We would like to thank the reviewer and added additional content to the discussion section of the manuscript regarding this topic.

To our knowledge, all current databases are incomplete regarding coverage of the tree of life, including NCBI Nt. FDA-ARGOS genomes are a subset of NCBI Nt. We agree with the reviewer that genomes in NCBI Nt are of variable quality and may contribute to uncertainties in data analysis as demonstrated in our updated use case 1.

We are not claiming superiority but propose control of uncertainties in data analysis through “walled off” genomes within NCBI Nt that passed through rigorous quality control fit for diagnostic use. The FDA-ARGOS genome subset contains quality-controlled reference genomes for diagnostic use with minimum metadata, high quality assemblies, high depth of raw sequence coverage from two independent sequencing platforms, orthogonal validation and de-novo sequence-based taxonomy verified by NCBI ANI.

We fully support the use of NCBI Nt as a reference database, but with caution. FDA-ARGOS is an additional resource, specifically tailored for diagnostic purposes. Our primary goal for FDA-ARGOS genomes is to provide a reference method to enable *in silico* validation and advance innovation.

The high-quality reference genome data and quality metrics are publicly available, so the community can develop FDA-ARGOS like quality genomes and deposit. It is also our intent to expand FDA-ARGOS with existing high quality public genome data.

4. A more direct comparison of FDA-ARGOS with “standard databases” would be if the authors took the curated, finished FDA-ARGOS genomes and simulated the incomplete and error-prone genomes that may exist in NCBI nt, and then used this modified FDA-ARGOS database for purposes of *in silico* comparison. This would be probably the best method to separate the effects of database size/completeness and database quality. It's unclear why the authors did not do so.

This is a great suggestion. Unfortunately, we are not aware of published NCBI nt error profile metrics that would allow systematic simulation of incomplete and error-prone genomes. Instead, we updated use case 1 and generated normalized NCBI Nt database instances that resemble FDA-ARGOS in composition and size and ran 2400 simulations to test effects of database quality (see response to comment 1 and 2).

We ran additional simulations for use case 1 with randomly “chopped up” FDA-ARGOS genomes. We did not observe a significant impact on species identification performance. This is expected given that most classification algorithms “chop up” genomes as part of their processing. We hypothesize that there is a dependency between size of contigs and species identification performance however this study is outside the scope of this paper.

5. It would be helpful to the readers to understand the justification for why the authors chose their threshold metrics for the FDA-ARGOS database (i.e. 95% coverage at 20X depth, at least 5 representative genomes in that species) as constituting “regulatory-grade” genomes.

Thank you for your comment. We expanded the discussion section and added our justification.

6. For the *E. avium* case, it would be useful to make clear in the figures about how many isolates of *E. avium* in NCBI Nt were remaining after removing SAMN04327393 versus how many isolates of *E. avium* were in FDA-ARGOS (n=4 is mentioned but only in line 486). The authors state that “..reads classified as *E. avium* ranged from an average 3829 and 840 when FDA-ARGOS genomes were added...compared to an average 29 and 0 reads when these genomes were absent...”. I find it hard to believe that there were only 29 reads identified using MegaBLAST (versus 3829 with FDA-ARGOS) if there were 4 genomes of *E. avium* in NCBI Nt, no matter how poor the quality. Furthermore, it is a little deceptive to state that *E. avium* reads were 840 for Kraken with FDA-ARGOS but 0 for Kraken because the standard Kraken database doesn't contain *E. avium* sequences. A more fair assessment, as mentioned previously, is to add the NCBI Nt *E. avium* sequences to the Kraken database and re-run the algorithm.

We revised use case 1 as described in our responses to comments 1 and 2. We utilized the 4 *E. avium* genome assemblies from Supplemental Table 3 for subset NCBI Nt simulations to match the 4 *E. avium* genome assemblies in FDA-ARGOS. These genomes stay fixed for all simulations with MegaBLAST and Kraken to ensure consistency. SAMN04327393 is an *E. avium* genome contained within FDA-ARGOS. This isolate was used to generate the isolate and metagenome use case data. Hence, we excluded it from the FDA-ARGOS database for use case 1 simulations to avoid self-referencing.

We randomly selected microbial genome assemblies from NCBI Nt or ARGOS to simulate 200 database instances and matched composition and size. For example, FDA-ARGOS may have more genome assemblies for a species than NCBI Nt (rare). Here, we randomly selected genome assemblies for that microbial species from ARGOS for ARGOS database instance simulations.

Below are sample simulation results of 100 MegaBLAST runs using 1,000,000 subsampled data from each triplicate sample (See Supplemental Tables 4, 4a, 4b, 5, 5a and 5b for all simulation results):

***E. avium* Metagenome MegaBLAST Simulations**

- We observed that *E. avium* is consistently called
 - NCBI nt: (min 18, max 109, median 44)
 - ARGOS: (min 36, max 104, median 43)
- Considering all *Enterococcus* genus reads
 - On average, NCBI Nt top hit is *E. faecium* (11.52%), followed by *E. hirae* (2.88%), *E. faecalis* (0.21%) and *E. avium* (0.03%)
 - On average, ARGOS top hit is *E. avium* (88.68%)
- Considering all mapped reads
 - On average, NCBI Nt top hit is *Vibrio vulnificus* (18.11%), followed by 32 species with >1%
 - Simulations with NCBI Nt revealed extremely variable results with calls all over the place burying *E. avium* (on average 62 reads, 0.03% of all mapped reads), unclear why this happens
 - Interestingly, several simulations with NCBI Nt revealed *E. hirae* at extremely high read counts (up to 90,000, 46.72% of all mapped reads) versus *E. avium* (up to 109, 0.06% of all mapped reads), resulting in *E. hirae* coming up as top hit leading to potentially false species calls
 - All simulations with ARGOS resulted in *E. avium* as the top hit (on average 60 reads, 88.68% of all mapped reads). *E. hirae*, *E. durans*, *E. faecium* and *E. faecalis* were included as near neighbors as were all other species represented within NCBI Nt.
- Clearly shows quality matters

***E. avium* Isolate MegaBLAST Simulations**

- We observed that *E. avium* is consistently called, on average
 - NCBI Nt: 651639 number of reads, 92.04% of all mapped reads
 - (min 297836 (42.07%), max 754219, median 669295)
 - ARGOS: 628420 number of reads, **97.89%** of all mapped reads
 - (min **544710 (84.85%)**, max 719035, median 629205)
- Considering all *Enterococcus* genus reads
 - On average, NCBI nt top hit is *E. avium* (92.04%), followed by *E. hirae* (3.10%), *E. faecium* (2.46%), *E. faecalis* (0.11%) and *E. durans* (0.04%)
 - On average, ARGOS top hit is *E. avium* (97.89%), followed by *E. faecium* (0.91%), *E. faecalis* (0.33%), *E. durans* (0.24%) and *E. hirae* (0.01%)
- Considering all mapped reads
 - On average, simulations with both NCBI Nt and ARGOS resulted in *E. avium* as the top hit
 - Interestingly, several simulations with NCBI Nt revealed *E. hirae* as top hit, a similar phenomenon seen with the *E. avium* metagenome data. This could potentially lead to false positive species identification calls.
- Again, shows quality matters.

We updated the discussion section and generated an updated Figure 4 to demonstrate the effect of database quality on species identification performance.

We thank the reviewer for these suggestions.

7. For the Ebola case study, why were only FDA-ARGOS EBOV genomes assessed? EBOV genomes are well represented in NCBI Nt especially due to the amount of sequencing done during the West African EBOV outbreak and the ongoing recent outbreaks in the DRC. Why are FDA-ARGOS EBOV sequences needed specifically? Why can't a representative subset of EBOV sequences from NCBI Nt be used for the C-RM approach? It is stated that "FDA-ARGOS alone was sufficient for in silico comparison". However, wouldn't a random selection of representative EBOV sequences (non FDA-ARGOS) from NCBI Nt also be "sufficient"?

We also think that ARGOS-like quality is sufficient for the C-RM approach. We are currently working on an open source tool utilizing FDA-ARGOS reference genome characteristics. More specifically, we are assessing genome quality (e.g. coverage, ANI, GC content, assembly size), genome continuity (e.g. N50, L50, number of contigs), taxonomy and metadata (e.g. species name, isolation source, submitter, orthogonal reference method) metrics to allow the community to qualify genomes for ARGOS deposition.

In the meantime, requests for genome validation and addition can be send to the FDA-ARGOS team FDA-ARGOS@fda.hhs.gov. We clarified this in the manuscript.

8. The C-RM approach is elegant but still requires a gold standard. This needs to be underscored in the discussion, as for many diseases, there may not be a gold standard clinical diagnostic test. A particularly relevant example is early acute Lyme disease (*Borrelia burgdorferi* PCR has low sensitivity; two-tiered antibody testing takes 2-4 weeks). This should be mentioned as a potential limitation of in silico validation using C-RM.

The proposed C-RM combines clinical validation of representative organisms against a reference method with dry lab testing of any number of desired organisms. The clinical validation can be done with any scientifically valid reference method. We updated this in Fig 1.

We also added a limitation in the discussion section to specifically address cases where orthogonal reference method testing is not feasible.

MINOR POINTS:

1. The authors mention using two different read technologies, PacBio and Illumina, in generating consensus sequences. Is this really needed? Arguably, the long reads may be needed for gene synteny but this would be primarily useful for unknown and/or novel reference genomes rather than genomes

for which there is at least one high- or moderate-quality reference in NCBI nt. Furthermore, the use of even PacBio and Illumina may not even be sufficient to finish genomes with highly repetitive genome structures such as *Prevotella copri*, as elegantly described by Bhatt, et al and colleagues (https://www.nature.com/articles/nbt.4266?WT.feed_name=subjects_genome-informatics), in which nanopore sequencing and de novo assembly were required. This should be mentioned somewhere in the manuscript.

We treat every genome as a novel reference genome. Based on current experiences with the FDA ARGOS effort, assembly quality appears to be species and technology dependent. For most species, PacBio was sufficient to generate the top assembly. For some species, PacBio + Illumina or Illumina only data generated the top assembly.

We agree with the reviewer that PacBio and Illumina data is likely not sufficient to finish genomes. The goal of ARGOS is to generate and make publicly available near finished genomes of sufficient quality for diagnostic purposes. The quality metrics were an unknown at the beginning of the effort and existing public genomes were mostly untraceable with missing minimum metadata and more importantly underlying raw data. We were not able to regenerate assemblies or attribute genomes.

The implemented and validated hybrid approach based on long and short read sequencing technology was the best strategy at the time. However, we found that PacBio may be sufficient to generate near finished genomes de novo. We are also looking to qualify other NGS technology data (e.g. Nanopore) but these efforts are outside the scope for this paper.

We added a clarification that the goal of FDA-ARGOS is to generate and make publicly available near finished genomes of sufficient quality for diagnostic purposes.

2. Regarding lines 360-365, it would be helpful for more description of what is meant by “gaps” in the database that FDA-ARGOS sought to address. Are these genomes? Near neighbors?

Generally, “gaps” were defined as non-existent traceable and attributable diagnostic relevant high-quality genomes. We clarified this in the manuscript.

3. The authors should mention that finishing the ends of viral genomes with 5'/3' RACE, while critically important in virology such as development of an infectious clone, is likely less important for the purposes of reference-based NGS alignment, as the ends of viruses are usually not well-represented in metagenomic sequencing (especially with library preparation techniques using transposons such as Nextera which cleave internal to the sequence).

We thank the reviewer for this insightful comment.

We agree that finishing ends of viral genomes with 5'/3' RACE may not be critical for viral reference-based NGS alignment. This is not a quality metric used for FDA-ARGOS inclusion as outlined in the paper. Finishing ends of viral genomes is a desired metric to obtain the highest quality viral reference genome.

FDA-ARGOS efforts encompass several collaborative efforts with different goals. As mentioned by the reviewer, finishing ends is important for other applications (therapeutics, vaccines). The FDA-ARGOS database resource is utilized for these efforts, but this is out of scope for this paper.

We clarified that finishing ends with RACE is optional in the viral reference genome sequencing and assembly method section.

4. It might be preferable to describe the case studies as 2 rather than 3, as the first case study appears to be embedded with the EBOV example.

We revised the manuscript to reflect 2 case studies.

Reviewer #2 (Remarks to the Author):

In their manuscript “FDA-ARGOS: A Public Quality-Controlled Genome Database Resource for Infectious Disease Sequencing Diagnostics and Regulatory Science Research”, Dr. Sichtig et al. describe extensive efforts to generate a publicly available database as a tool to support innovation of emerging technologies and provide an in-silico comparator tool that could reduce the burden for completing ID-NGS clinical trials. The manuscript also discusses quality control metrics for the proposed FDA-ARGOS database and presents three examples demonstrating potential applications for FDA-ARGOS in infectious disease diagnostics.

Major Comments

1) This reviewer congratulates the authors for undertaking a very important effort in generating a reference database of regulatory-grade, high-quality genome sequences. In several of the discussed applications of ID NGS, it will be common to generate sequencing data for microbes and pathogens that have no identical reference genome available (as the full genomic diversity of bacteria, viruses, and eukaryotic pathogens is not known and continuously changing). While likely not of higher quality, test-specific databases may be larger than the FDA-ARGOS database. It is therefore possible that test-specific databases will contain one or more reference sequences that are more similar to a given organism identified by a new test. In this case, analysis of the same raw sequencing data with the FDA-ARGOS database may produce different (but not necessarily wrong) results. The authors should consider discussing the limitations of inherently incomplete databases and potential pathways to resolving discrepancies that result from varying database completeness.

We thank the reviewer for the remarks. We clarified the limitations of inherently incomplete databases in the discussion section. In addition, we emphasized our invitation to the community to contribute to the ARGOS database by sample or genome submission, or crowd sourcing existing high-quality genomes for inclusion based on provided quality criteria.

The challenges of species and subspecies calling, generating a complete microbial tree of life database and species-specific variant databases are out of scope for this manuscript. However, there are several

FDA-ARGOS collaborative efforts that focus on generating quasi-species reference genome data to support vaccine and therapeutics development (e.g. Ebola 7U vs 8U detection).

The goal of ARGOS is to generate high-quality subsets of near finished reference genomes for valuable diagnostic microbial species. The FDA-ARGOS genome subset contains quality-controlled reference genomes for diagnostic use with minimum metadata, high quality assemblies, high depth of raw sequence coverage from two independent sequencing platforms, orthogonal validation and de-novo sequence-based taxonomy verified by NCBI ANI.

We revised use case 1 and demonstrated that genome quality matters. Selection of diverse isolates from microbial species to generate representative near finished genomes is generally sufficient for most genus and species reference-based ID NGS alignment applications.

We agree with the reviewer that reference genome generation is an ongoing process.

Clearly, scalability of efforts required to create curated and comprehensive databases is a concern. The authors briefly discuss how scalability of the FDA-ARGOS database could be increased (lines 589-592). Could the authors expand on the strategy and timeline for including existing genomic data into a regulatory-grade database? What quality control metrics could be implemented to include existing (rather than newly-generated) genome data into the FDA-ARGOS database?

We thank the reviewer for the comments and opportunity to clarify expansion efforts for FDA-ARGOS.

We agree that scalability and sustainability are important factors for this effort. We expanded the scalability of ARGOS section with more specific information on proposed quality metrics that will be used for external genome qualification: genome quality (e.g. coverage, ANI, GC content, assembly size), genome continuity (e.g. N50, L50, number of contigs), taxonomy and metadata (e.g. species name, isolation source, submitter, orthogonal reference method). The introduction and detailed description of the open-source qualification tool will be covered in another manuscript and is out of scope for this paper.

We also added the option that the community can contact us at FDAARGOS@fda.hhs.gov for latest updates on tool development and submit requests to add existing genome data to FDA-ARGOS. Currently, there is a unique opportunity to submit samples for FDA-ARGOS genome generation and automatic inclusion.

2) Lines 368-369: “An initial collection criterion focused on sequencing at least 5 diverse isolates per species”

The authors may want to discuss how intra-species diversity may affect completeness of genetic representation by 5 isolates. This number may be too small for some taxa, especially bacteria with high genetic diversity and viruses.

We agree with the reviewer and revised our discussion section to address the ‘at least 5 representative genome’ metric.

3) Use Case 1

This contrived sample contained ~4000 sequencing reads of *E. avium*, representing approximately 10-15% of the bacterial genome. Do the authors consider this a sufficient representation for a positive result? This reviewer does not necessarily disagree with a positive interpretation. Given the intended use of the FDA ARGOS database, it would be valuable to the audience to learn more about how the authors think about defining true positive results.

We revised use case 1 to address the goal of this manuscript to show impact of reference database quality on result calling. The number of mapped *E. avium* reads for the isolate and metagenome samples are an estimation from subsampling 1Mio reads. The percent mapped reads reflect sample composition better.

The threshold determination for positivity calling is expected to be ID NGS application dependent. Use case 1 showed the challenges with *E. avium* calling, especially for the metagenome sample. We clarified the determination for positive calls for the Ebola use case 2 in the methods section.

As mentioned previously, the challenge of species and subspecies calling for reference-based ID NGS applications is outside the scope of this paper.

Minor Comments

- Lines 297-298, 312-313: “For this study, the taxon associated with the first reported alignment was used as the taxonomic label for each read...”

Albeit easy to implement, this approach oversimplifies the complexity of the classification problem. For example, ‘assigning the taxonomic label for each read to the first reported alignment’ is highly dependent on the accuracy of reference sequence annotations, which are frequently incorrect or incomplete in the NCBI nt database. In addition, sequencing reads may map equally well to reference sequences with different taxonomic annotations. In this case, simply assigning the sequencing read to the ‘first reported alignment’ (presumably sorted by max alignment score).

We agree with the reviewer that sophisticated algorithms and curated databases are needed for diagnostic purposes. In addition, subject matter expertise may be needed to interpret reference-based ID NGS alignment output. Not everyone has access to these tools and expertise. Therefore, we selected publicly available tools, MegaBLAST and Kraken, to demonstrate the impact of reference genome quality on mapped reads output.

We revised our method description to include this limitation. We clarified that the taxon associated with the first reported alignment, sorted by max alignment score, was used as the taxonomic label for each read. Original MegaBLAST results were summarized to report the number of reads associated with each unique NCBI taxonomy ID called.

- Lines 249-250: “...paired end reads were trimmed utilizing a quality of 0.05 and reads below 50bp in length were removed...”

Please provide additional details for quality trimming.

We would like to thank the reviewer for the opportunity to clarify our trimming methods.

Sequencing reads are trimmed on CLC using a modified-Mott trimming algorithm. Quality scores in CLC are on a Phred scale and Phred quality scores (Q) are defined as $Q = -10 \log_{10}(P)$ where P is base calling error probability. The first step in the process converts the quality score to error probability using $p(\text{error}) = 10^{-(Q/10)}$ and then calculates new values for each base using the equation $\text{Limit-}p(\text{error})$. This limit can be adjusted depending on stringency with high numbers represent less stringent cutoffs. CLC Workbench then calculates the running sum of this value across the length of the sequence. If the sum drops below zero, it is set to zero and the region ending at the highest value of the running sum starting at the last zero will not be trimmed.

Qiagen. (2019) CLC bio manuals. Retrieved from http://resources.qiagenbioinformatics.com/manuals/clcgenomicsworkbench/803/index.php?manual=Quality_trimming.html

- Lines 250-253: “Trimmed reads were then mapped to E. avium assembly GCF_000407245.1 and H. sapiens assembly GCA_000001405.27. Mapping parameters were as follows: mismatch costs=2, insertions costs=3, deletion costs=3, length and similarity fraction = 0.8.”

What mapping tool was used for this analysis?

We used the CLC genomics workbench v 10.1.1 mapping function for our analysis. This analysis was done with a local alignment of the reads to the reference.

- Line 262: “The threshold for positive calls was determined by the no template control (NTC).” This sentence is not entirely clear. Was the number of reads obtained with the NTC defined as the threshold for a positive call? If so, the use of an NTC for this purpose may provide a false sense of security as higher read counts may be obtained in samples positive for related viruses (causing false-positive results).

We would like to thank the reviewer for their comments.

For highly multiplexed sequencing, small indices were attached to each sample so that each sequencing read captured can later be attributed to its source. Sequencing errors of these indices can result in sequencing reads being mis-binned during deconvolution. Unfortunately, there is no method which deals with these issues 100% effectively.

To counteract this mis-binning we devoted sequencing space on each run to three non-template control samples. A cutoff is calculated for each reference organism, using these non-template controls, as the average plus three times the standard deviation. This method defines a background level of sequencing read “bleed” caused by the sequencing platform and was used for MIPS analysis

in a previously published paper (Koehler, Hall et al. 2014). We have clarified how we defined this cutoff in the paper.

Samples which have large numbers of reads attributed to related viruses are more likely to result from low stringency read mapping settings and should be followed up with a secondary analysis such as de novo assembly and reference-based mapping.

- Lines 360-362: “FDA, DOD, NCBI and other agencies using scientific literature, a phylogenetic data mining approach, and FDA microbial species-specific guidance documents identified more than 1000 gaps in public microbial genomic repositories. We prioritized these gaps and selected biothreat microorganisms, common clinical pathogens and closely related species (See Supplemental Materials for the organism gap list).”

Which supplemental file are the authors referring to?

Supplemental Materials Word Document under Additional Files (FDA-ARGOS Wanted Organism List). We added this information to the manuscript.

- Lines 449-451 “For frame-of-reference, we would need over 30,000 reads to de novo assemble an entire genome of approximately 5 Mb at 1X coverage, assuming a read size of 150 bp and perfect quality of each generated read at all positions”

The authors may want to choose a more realistic example. De novo assembly of a bacterial genome is not possible with 1X coverage.

We updated the hypothetical example using 20X coverage for a more realistic illustration.

- Lines 465-467 “Interestingly, while *E. avium* genomes were available in the NCBI nt database and part of the read classification for MegaBLAST analyses, positive ID-NGS identification required the addition of quality-controlled FDA-ARGOS reference genomes”

Did the authors assess whether the *E. avium* genomes were available in the NCBI nt database were complete? Were they correctly identified? What was the average nucleotide identity of SAMN04327393 with the *E. avium* genomes contained in the NCBI nt database? Please provide additional information for the difference in results.

We significantly revised use case 1 for a more fair comparison. Information on NCBI Nt and FDA-ARGOS *E. avium* genome statistics is available in supplemental tables 1 and 3. We utilized the 4 *E. avium* genome assemblies from Supplemental Table 3 for subset NCBI Nt simulations to match the 4 *E. avium* genome assemblies in FDA-ARGOS. These genomes stay fixed for all simulations with MegaBLAST and Kraken to ensure consistency. SAMN04327393 is an *E. avium* genome contained within FDA-ARGOS. This isolate was used to generate the isolate and metagenome use case data.

Hence, we excluded it from the FDA-ARGOS database for use case 1 simulations to avoid self-referencing.

- Lines 488-489 “These top hits were potentially database contaminants and illustrate the risk of using non-curated databases in ID-NGS diagnostics”

In addition, the authors may want to consider discussing the potential impact of different classification algorithms or criteria.

Here we showed performance of two classification algorithms (MegaBLAST, Kraken) and 200 NCBI Nt and FDA-ARGOS database instances. The goal of use case 1 was to show the impact of reference genome quality on classification algorithm output. A direct method comparison study is outside the scope for this paper. We added information regarding the impact of different classification algorithms and criteria to the discussion section.

- Lines 614-615 “Experiments and data presented here support the latter of these two arguments.”

This reviewer does not disagree with this conclusion for the given examples. However, this was in part because the FDA ARGOS database included reference genomes for the studied examples. The authors should discuss how results may differ in cases where a high-quality genome is not available for a pathogen contained in a given sample.

We thank the reviewer for the comment.

We revised our use case 1 to address the impact of genome quality on performance of classification algorithms. We believe that these new simulations provide a more fair comparison. Our results clearly demonstrated that genome quality critically impacts final call performance.

For example, the varied quality of NCBI Nt reference genomes significantly impacted the classification results, burying *E. avium* reads for the metagenome sample cases (Supplemental Table 4, 4a, 4b).

Reviewers' Comments:

Reviewer #1:

Remarks to the Author:

In this revised manuscript, Sichtig and colleagues address many of the points that I and the other reviewer raised in the initial review. However, I still have a few concerns, particularly about revised use case 1. In my opinion, it is critical to precisely define the comparisons being made, as the main conclusions of the paper depend on this.

1. I agree with and appreciate the additional work put into generating the 200 randomized database instances using for each instance "the identical species composition and number of assemblies per species". I carefully reviewed the results, including Supplemental Tables 4, 4a, 4b, 5, 5a, and 5b, and the new updated Figure 4 heat map.

a. The results are confusing. The metagenomic data that you show in Supplemental Table 4 look "too bad". For instance, I find it hard to believe that there are an average of 34,879 reads to *Vibrio vulnificus* and 17,108 reads to *E. coli* using the normalized NCBI nt database with MegaBLAST but 0 reads to these 2 organisms using the normalized FDA-ARGOS database with MetaBLAST using the same parameters and cutoffs (and assuming that the normalized FDA-ARGOS database has the same level of genome representation of *V. vulnificus* and *E. coli*, as implied by your description of the database construction). Actually, I note that there is sometimes 1 read mapping to *E. coli*/*V. vulnificus* with the FDA-ARGOS database, but I am making the assumption that this was averaged to zero. An explanation needs to be given on why we see 34,879 reads to *V. vulnificus* and 17,108 reads to *E. coli* in the metagenomic dataset with the normalized NCBI nt database but 0 and 0 reads with the normalized FDA-ARGOS dataset. Also, some of the bacteria that I see in the normalized NCBI list correspond to well-known metagenomic sequencing contaminants, such as *Pseudomonas protegens/putida* and *Bacillus* sp. (see Salter, et al., <https://bmcbiol.biomedcentral.com/articles/10.1196/s12915-014-0087-z>). In short, the metagenomic analysis done using the normalized NCBI nt database using raw MegaBLAST alignment (importantly) without any post-processing taxonomic classification or filtering looks far more believable to me than the FDA-ARGOS results. I can postulate some potential explanations as follows: (1) are the reads aligning to other bacteria aligning to plasmid sequences that are not part of the core genome, (2) are different stringency thresholds being applied in MegaBLAST for the normalized NCBI nt and FDA-ARGOS databases, (3) does this represent a failure of alignment and/or classification (for KRAKEN), (4) are these reads repeat and/or low-quality sequences in the human genome. In summary, whether or not the *V. vulnificus*, *E. coli*, *P. putida/protegens*, *Bacillus*, etc. reads identified using the annotated NCBI nt database are "real" needs to be determined and described in detail. In my opinion, this discrepancy in results is so vast that it cannot be explained by just minor differences in sequencing quality.

2. One possibility is that these additional hits to bacteria are due to "human" or other contamination of the genome sequences in NCBI nt. For this, I note that metagenomic analysis typically involves a "human host computational subtraction" step in which reads to background human sequences are removed by alignment to hg38. Was this done for this analysis? This may help clean up the data. Either way, an explanation needs to be given on why there only ~'60 reads to *Enterococcus avium* when using FDA-ARGOS and 34,870 reads to *V. vulnificus*, 17,108 reads to *E. coli*, etc. The discrepancy is worrisome especially since the base metagenome data are identical.

3. Is the *E. avium* metagenome that was generated a DNA only metagenome or a combined DNA/cDNA (with reverse transcription) metagenome? If the latter, there may be many more hits to extraneous bacteria because of the preponderance of highly conserved bacterial 16S ribosomal RNA (rRNA) sequences.

4. One additional critique in the method used is that although there were "the same number of

randomly chosen *E. avium*, *E. faecium*, etc. reference genomes, the vast additional sequences available in NCBI nt were presumably retained. The results from “2,400 simulations” appear to result in more “precise” species-specific calls simply because there is less coverage of bacterial genomes overall in the FDA-ARGOS database related to NCBI nt. With large databases, more stringent filtering and taxonomic classification criteria, and/or host subtraction methods needs to be applied. A fairer composition would have been for the 2,400 simulation to choose an identical number of genomes in total for the normalized NCBI Nt and FDA-ARGOS database rather than choose the same number of *Enterococcus* genomes only.

5. I note that *E. avium* had slightly higher average read number with the normalized NCBI nt database rather than FDA-ARGOS. Thus, *E. avium* detection was more sensitive. An explanation needs to be provided for why 20% of total reads mapped to other microbial species. I would agree that this would be a dilemma for species identification applications. The number of 20% is staggeringly high; all metagenomic pipelines that I am aware of that use the NCBI nt database get much lower hits to other microbial species and – if proper filtering, taxonomic classification, and host subtraction are performed -- they almost always represent known contaminants in the database such as *Pseudomonas protegens*, *Bradyrhizobium* sp., *Ralstonia* sp., *Delftia* sp., etc. rather than pathogens such as *V. vulnificus*.

6. An explanation is not given as to why some simulations with the normalized NCBI Nt database showed *Enterococcus hirae* as the top hit. This should be investigated – is this due to a misassignment of one or more *Enterococcus avium* genomes as *Enterococcus hirae* in GenBank nt (a database issue) or another problem?

7. The authors should comment on the viability and practical aspects of using both FDA-ARGOS and NCBI Nt simultaneously in the manuscript. Expansion of FDA-ARGOS for completeness (i.e. viruses, fungi, and parasites) may take years, especially for eukaryotic genomes, which are extremely large in size.

8. With regards to the EBOV genomes, I would like to re-iterate the point that having a large number (>10,000) of viral genomes that capture the diversity of RNA viruses is preferable to a few limited number of genomes, as in the FDA-ARGOS database. True, these genomes are presumably of lower quality but there are vastly more of them. This needs to be discussed. I do not see the need for metagenomic identification purposes of having a viral FDA-ARGOS database, for instance, whereas your data does appear to show that higher quality may result in a more precise identification with genomes that are closely related (i.e *Enterococcus avium* versus *Enterococcus hirae*). Nevertheless, for viruses, capturing sequence diversity with more genomes appears to be important. A relevant example would be the identification of Ekpoma virus (<https://journals.plos.org/plosntds/article?id=10.1371/journal.pntd.0003631>), a novel rhabdovirus virus in African individuals, which would not had been possible had the genome of Bas-Congo rhabdovirus (<https://journals.plos.org/plospathogens/article?id=10.1371/journal.ppat.1002924>) not been already in the database. In short, diversity, size, and completeness appear to trump accuracy, at least for viral databases. I am wondering if the authors can comment on this.

Reviewer #2:

Remarks to the Author:

This reviewer thanks the authors for addressing their comments and concerns.

Reviewer's comments:

Note: We highlighted all edits in response to reviewer comments throughout the paper.

Reviewer #1 (Remarks to the Author):

In this revised manuscript, Sichtig and colleagues address many of the points that I and the other reviewer raised in the initial review. However, I still have a few concerns, particularly about revised use case 1. In my opinion, it is critical to precisely define the comparisons being made, as the main conclusions of the paper depend on this.

I agree with and appreciate the additional work put into generating the 200 randomized database instances using for each instance "the identical species composition and number of assemblies per species". I carefully reviewed the results, including Supplemental Tables 4, 4a, 4b, 5, 5a, and 5b, and the new updated Figure 4 heat map.

We thank the reviewer for the remarks and the opportunity to address additional concerns about revised use case 1.

1. The results are confusing. The metagenomic data that you show in Supplemental Table 4 look "too bad". For instance, I find it hard to believe that there are an average of 34,879 reads to *Vibrio vulnificus* and 17,108 reads to *E. coli* using the normalized NCBI nt database with MegaBLAST but 0 reads to these 2 organisms using the normalized FDA-ARGOS database with MetaBLAST using the same parameters and cutoffs (and assuming that the normalized FDA-ARGOS database has the same level of genome representation of *V. vulnificus* and *E. coli*, as implied by your description of the database construction). Actually, I note that there is sometimes 1 read mapping to *E. coli*/*V. vulnificus* with the FDA-ARGOS database, but I am making the assumption that this was averaged to zero. An explanation needs to be given on why we see 34,879 reads to *V. vulnificus* and 17,108 reads to *E. coli* in the metagenomic dataset with the normalized NCBI nt database but 0 and 0 reads with the normalized FDA-ARGOS dataset. Also, some of the bacteria that I see in the normalized NCBI list correspond to well-known metagenomic sequencing contaminants, such as *Pseudomonas protegens/putida* and *Bacillus* sp. (see Salter, et al., <https://bmcbiol.biomedcentral.com/articles/10.1196/s12915-014-0087-z>). In short, the metagenomic analysis done using the normalized NCBI nt database using raw MegaBLAST alignment (importantly) without any post-processing taxonomic classification or filtering looks far more believable to me than the FDA-ARGOS results. I can postulate some potential explanations as follows: (1) are the reads aligning to other bacteria aligning to plasmid sequences that are not part of the core genome, (2) are different stringency thresholds being applied in MegaBLAST for the normalized NCBI nt and FDA-ARGOS databases, (3) does this represent a failure of alignment and/or classification (for KRAKEN), (4) are these reads repeat and/or low-quality sequences in the human genome. In summary, whether or not the *V. vulnificus*, *E. coli*, *P. putida/protegens*, *Bacillus*, etc. reads identified using the annotated NCBI nt database are "real" needs to be determined and described in detail. In my opinion, this discrepancy in results is so vast that it cannot be explained by just minor differences in sequencing quality.

We further analyzed the results from Supplemental Table 4. Specifically, we investigated the top species mapped reads from the normalized NCBI Nt database instance runs with the MegaBLAST tool (Table 4b contains the raw data for all runs). We confirm that the MegaBLAST tool was used with the same parameters, cutoffs and an identical number of randomly selected representative species assemblies from NCBI Nt and FDA-ARGOS. We clarified that an identical number of assemblies per species were used in the methods section describing the database construction. Line 456 already stated that an identical number of assemblies per species were used.

In summary, results showed mislabeling due to human contaminants in randomly selected microbial genomes from NCBI Nt database instances. Therefore, we see *Vibrio vulnificus* at an average 34,879 mapped reads and *E.coli* at an average 17,108 mapped reads. FDA-ARGOS genomes for these species did not show human contamination.

For the additional analysis, we selected all unique read hits from 5 randomly selected NCBI Nt database instances for each of the top 5 microbial species hits from Supplemental Table 4, including *Vibrio vulnificus* and *E. coli*. The table below shows NCBI Nt database instance selected, total unique read hits for that species, number of all accessions and number of all accessions that hit to human.

SPECIES NAME	DATABASE INSTANCE	TOTAL UNIQUE READ HITS	#ACCESSIONS	#ACCESSIONS HIT TO HUMAN
Vibrio vulnificus	NCBI_13	228,734	19	16
	NCBI_22	239,570	19	16
	NCBI_87	193,245	7	6
	NCBI_84	275,470	24	22
	NCBI_74	116,930	6	6
Edwardsiella tarda	NCBI_9	82,382	9	8
	NCBI_10	112,074	9	8
	NCBI_20	119,576	9	8
	NCBI_40	117,541	9	8
	NCBI_43	105,282	9	8
Escherichia coli	NCBI_0	78,978	9	8
	NCBI_3	11,594	2	2
	NCBI_5	57,021	2	1
	NCBI_7	36,805	2	2
	NCBI_8	73,966	6	2
Enterococcus faecium	NCBI_1	54,848	13	9
	NCBI_3	12,186	3	2
	NCBI_4	43,175	3	1
	NCBI_5	46,443	2	1
	NCBI_21	24,178	1	1
Klebsiella aerogenes	NCBI_3	179,373	3	3
	NCBI_11	59,750	6	4
	NCBI_23	84,381	3	3
	NCBI_25	190,812	3	3
	NCBI_28	84,438	3	3

We clarified this in the manuscript.

2. One possibility is that these additional hits to bacteria are due to “human” or other contamination of the genome sequences in NCBI nt. For this, I note that metagenomic analysis typically involves a “human host computational subtraction” step in which reads to background human sequences are removed by alignment to hg38. Was this done for this analysis? This may help clean up the data. Either way, an explanation needs to be given on why there only ~60 reads to *Enterococcus avium* when using FDA-ARGOS and 34,870 reads to *V. vulnificus*, 17,108 reads to *E. coli*, etc. The discrepancy is worrisome especially since the base metagenome data are identical.

See above. We agree that this is worrisome. Recommendations for rigorous quality control, including human and lab contaminant screening was added to the manuscript.

Human host computational subtraction was not performed for our metagenome analysis with NCBI Nt database instances and the MegaBLAST or Kraken tool as this database is not design to benefit just tailored applications. Not all bioinformatic pipelines include a de-hosting step.

3. Is the *E. avium* metagenome that was generated a DNA only metagenome or a combined DNA/cDNA (with reverse transcription) metagenome? If the latter, there may be many more hits to extraneous bacteria because of the preponderance of highly conserved bacterial 16S ribosomal RNA (rRNA) sequences.

DNA only. We clarified this in the manuscript.

4. One additional critique in the method used is that although there were “the same number of randomly chosen *E. avium*, *E. faecium*, etc. reference genomes, the vast additional sequences available in NCBI nt were presumably retained. The results from “2,400 simulations” appear to result in more “precise” species-specific calls simply because there is less coverage of bacterial genomes overall in the FDA-ARGOS database related to NCBI nt. With large databases, more stringent filtering and taxonomic classification criteria, and/or host subtraction methods needs to be applied. A fairer composition would have been for the 2,400 simulation to choose an identical number of genomes in total for the normalized NCBI Nt and FDA-ARGOS database rather than choose the same number of *Enterococcus* genomes only.

See 1. above. All instances of normalized NCBI nt and FDA-ARGOS contain an identical number of genomes. Additional sequences were not retained in the normalized NCBI Nt database instances.

5. I note that *E. avium* had slightly higher average read number with the normalized NCBI nt database rather than FDA-ARGOS. Thus, *E. avium* detection was more sensitive. An explanation needs to be provided for why 20% of total reads mapped to other microbial species. I would agree that this would be a dilemma for species identification applications. The number of 20% is staggeringly high; all

metagenomic pipelines that I am aware of that use the NCBI nt database get much lower hits to other microbial species and – if proper filtering, taxonomic classification, and host subtraction are performed - they almost always represent known contaminants in the database such as *Pseudomonas protegens*, *Bradyrhizobium* sp., *Ralstonia* sp., *Delftia* sp., etc. rather than pathogens such as *V. vulnificus*.

See 1. above. We agree that quality control of microbial reference databases is necessary, especially for diagnostic use. Use case 1 demonstrated that lack of quality-controlled reference genomes challenged the accuracy of reference-based ID-NGS alignment applications. This is about a diagnostic answer not about a researcher or expert user interpreting results and suggesting which of the top hits are possible contaminants and which are potential pathogens.

6. An explanation is not given as to why some simulations with the normalized NCBI Nt database showed *Enterococcus hirae* as the top hit. This should be investigated – is this due to a misassignment of one or more *Enterococcus avium* genomes as *Enterococcus hirae* in GenBank nt (a database issue) or another problem?

We thank the reviewer for this insightful comment.

Additional analysis of results revealed that whenever SAMN03198084 (*Enterococcus hirae*) was randomly selected as part of the normalized NCBI Nt database, *Enterococcus hirae* was shown as the top hit. We performed pairwise ANI calculations on *Enterococcus hirae* and *Enterococcus avium* genomes contained within the normalized NCBI NT database instances using the OrthoANI algorithm (<https://www.ncbi.nlm.nih.gov/pubmed/26585518>). The calculated ANI score for SAMN03198084 showed higher correlation to the *Enterococcus avium* genomes than the remaining *Enterococcus hirae* genomes. Therefore, *Enterococcus hirae* is the top hit for several simulations. We clarified this database issue in the manuscript.

7. The authors should comment on the viability and practical aspects of using both FDA-ARGOS and NCBI Nt simultaneously in the manuscript. Expansion of FDA-ARGOS for completeness (i.e. viruses, fungi, and parasites) may take years, especially for eukaryotic genomes, which are extremely large in size.

We created the normalized NCBI Nt database instances with identical coverage of reference genomes for fair comparison. See 1 above. The entire NCBI Nt database has more coverage but also contains even more varying levels of quality genomes. Practical use of both databases depends on specific application and use (*in silico* validation tool vs reference database). This paper focused on reference genomes for diagnostic use.

The revised manuscript discussion section states that the FDA-ARGOS reference genome resource is a constantly evolving public database intended to mature over time with community support and genomic technology advancements. The authors agree, as stated in the manuscript, continued population and expansion of the FDA-ARGOS database resource will be required to cover the panoply of infectious microorganisms. We believe further population and curation of the database will support the success of FDA-ARGOS and promote adoption by the NGS community.

Furthermore, we fully support the use of NCBI Nt as a reference database, but with appropriate controls to mitigate the issues addressed in this manuscript. Again, the dichotomy of application between pathogen discovery and diagnostics appears to be the root of this comment. The establishment of FDA-ARGOS within NCBI provides an additional resource specifically tailored for diagnostic purposes whereas the use of NCBI Nt is more relevant for use when searching for completely novel pathogens.

8. With regards to the EBOV genomes, I would like to re-iterate the point that having a large number (>10,000) of viral genomes that capture the diversity of RNA viruses is preferable to a few limited number of genomes, as in the FDA-ARGOS database. True, these genomes are presumably of lower quality but there are vastly more of them. This needs to be discussed. I do not see the need for metagenomic identification purposes of having a viral FDA-ARGOS database, for instance, whereas your data does appear to show that higher quality may result in a more precise identification with genomes that are closely related (i.e *Enterococcus avium* versus *Enterococcus hirae*). Nevertheless, for viruses, capturing sequence diversity with more genomes appears to be important. A relevant example would be the identification of Ekpoma virus (<https://journals.plos.org/plosntds/article?id=10.1371/journal.pntd.0003631>), a novel rhabdovirus virus in African individuals, which would not had been possible had the genome of Bas-Congo rhabdovirus (<https://journals.plos.org/plospathogens/article?id=10.1371/journal.ppat.1002924>) not been already in the database. In short, diversity, size, and completeness appear to trump accuracy, at least for viral databases. I am wondering if the authors can comment on this.

We thank the reviewer for the remarks.

We agree that diversity, size and completeness are important factors for pathogen discovery applications. The more 'data hints' available the better. In these research cases, the entire NCBI Nt database would be a better resource; however, for diagnostics, accuracy trumps diversity with poor accuracy. An anecdotal example from authors capturing this difference in application stems from the 2014 West African outbreak for EBOV Makona where samples for this database originated. In the outbreak, patients were triaged into the EVD ward versus the non-EVD based on one diagnostic real-time PCR test. Multiple different platforms were available for use including sequencing using NCBI Nt as a reference database; however only the real time PCR was used. The reason for this was the magnitude of the diagnostic answer. Positive for EBOV led to the patient being housed in the EVD ward while a negative led to the non. Minimizing false positives (putting non-EVD patients in with EVD patients) and false negatives (contaminating the EVD ward) was more important than catching perhaps 1 in 1 million chance there is a novel pathogen causing similar symptoms in these patients.

The focus of our manuscript are quality-controlled reference genomes for diagnostic and regulatory science use. We demonstrated in the manuscript that quality of genomes impacts accuracy of reference-based ID-NGS alignment for queryable microbial pathogens, including viruses.

Furthermore, accuracy played an important role for FDA-ARGOS EBOV genomes generated from Public Health Agency Canada and Public Health England challenge materials. Specific positional coverage was important for frequency calculation of the 7U versus 8U content, potentially revealing impact of Ebola virus adaptation.

<https://www.ncbi.nlm.nih.gov/pmc/articles/PMC39657/>

<https://www.ncbi.nlm.nih.gov/pubmed/25214632>

In addition, coverage of each position of the genome and accuracy are expected to be critical factors for CRIPR-Cas9 applications.

Reviewer #2 (Remarks to the Author):

This reviewer thanks the authors for addressing their comments and concerns.

Reviewers' Comments:

Reviewer #1:

Remarks to the Author:

In this revised manuscript, Sichtig and colleagues have addressed in detail the remaining concerns that I have raised regarding the high number of spurious hits to *V. vulnificus* and *E. coli*, *Enterococcus hirae* finding, and importance of Ebola virus accuracy, and have also made appropriate edits to the manuscript. I thank the authors for addressing these comments, and believe that this work will be an important contribution to the field of metagenomic-based diagnostics.